

# An assessment of equatorial Atlantic interannual variability in OMIP simulations

Arthur Prigent[1] and Riccardo Farneti[1]

[1]Earth System Physics, The Abdus Salam International Centre for Theoretical Physics (ICTP), Trieste, 34134, Italy

**Correspondence:** Arthur Prigent (aprigent@ictp.it)

**Abstract.** The eastern equatorial Atlantic (EEA) seasonal cycle and interannual variability strongly influence the climate of the surrounding continents. It is thus crucial that models used in both climate predictions and future climate projections are able to simulate them accurately. In that context, the EEA seasonal cycle and interannual variability are evaluated over the period 1985-2004 in models participating to the Ocean Model Intercomparison Project Phases 1 and 2 (OMIP1 and OMIP2). The

main difference between OMIP1 and OMIP2 simulations is their atmospheric forcing: CORE-II and JRA55-do, respectively. Seasonal cycles of the equatorial Atlantic zonal winds, sea level anomaly and sea surface temperature in OMIP1 and OMIP2 are comparable to reanalysis datasets. Yet, some discrepancies exist in both OMIP ensembles: the thermocline is too diffusive and there is a lack of cooling during the development of the Atlantic cold tongue. In addition, the vertical ocean velocity in the eastern equatorial Atlantic in boreal summer is larger in OMIP1 than in OMIP2 simulations. The EEA interannual sea surface

temperature variability in the OMIP1 ensemble mean is found to be 51% larger ($0.62 \pm 0.04$ °C) than the OMIP2 ensemble mean ($0.41 \pm 0.03$ °C). Sensitivity experiments demonstrate that the discrepancy in interannual sea surface temperature variability between OMIP1 and OMIP2 is mainly attributed to their wind forcing. While the April-May-June zonal wind variability in the western equatorial Atlantic is similar in both forcing, the zonal wind variability peaks in April for JRA55-do and in May for CORE-II. Differences in surface heat fluxes between the atmospheric forcing datasets have no significant impacts on the

simulated interannual SST variability.

## 1 Introduction

Various regions of the globe are marked with large sea surface temperature (SST) variability (defined as the standard deviation of the monthly mean SST anomalies, Figure 1). In the extratropical regions, high SST variability (exceeding 1.2 °C) is observed in areas with strong SST gradients such as the Gulf Stream and Malvinas Current in the Atlantic Ocean, the Agulhas Current in

the Indian Ocean, and the Kuroshio Current in the Pacific Ocean (Deser et al. (2009), Figure 1a). Within the tropics, high SST variability occurs at the interannual timescale in the equatorial Pacific, mainly driven by El Niño/Southern Oscillation (ENSO), and in the equatorial Atlantic, driven by the Atlantic zonal and meridional modes. The simulation of SST variability by state-of-the-art ocean general circulation models (OGCMs) participating to the Ocean model intercomparison Project (OMIP) Phases 1 and 2 are shown in Figures 1b and c, respectively. In comparison to the OMIP1 ensemble (Figure 1b), the OMIP2 ensemble

mean (Figure 1c) depicts higher SST variability in eddy-rich regions like the Gulf Stream, Kuroshio, Malvinas and Agulhas



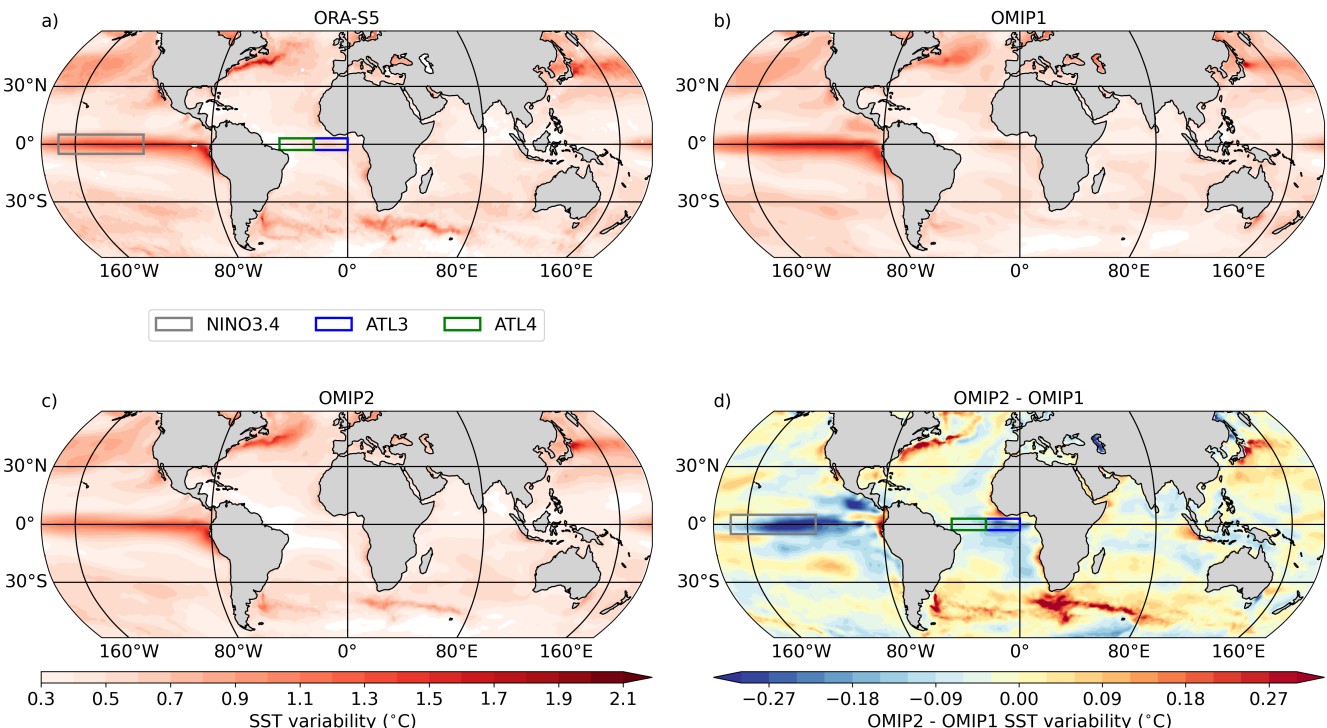

**Figure 1.** Standard deviation of SST anomalies for (a) ORA-S5, (b) the OMIP1 ensemble mean, (c) the OMIP2 ensemble mean spanning from January 1985 to December 2004. (d) Difference between the OMIP2 ensemble mean minus the OMIP1 ensemble mean. The grey, blue, and green boxes represent the NINO3.4 (170°W-120°W, 5°S-5°N), ATL3 (20°W-0°E, 3°S-3°N), and ATL4 (40°W-20°W, 3°S-3°N) regions, respectively.

current as well as in eastern boundary upwelling systems, most notably in the Angola-Benguela Area and off the coasts of Peru and Dakar (Figure 1d). Conversely, in the equatorial Pacific and Atlantic Oceans, the OMIP1 ensemble mean exhibits higher SST variability than the OMIP2 ensemble mean (Figure 1d). This study specifically addresses the discrepancies in the representation of interannual SST variability in the equatorial Atlantic between the OMIP1 and OMIP2 ensembles.

The SST in the equatorial Atlantic exhibits a marked seasonal cycle closely related to the seasonal displacement of the intertropical convergence zone (ITCZ). In March-April-May (MAM), highest temperatures are observed in the equatorial region (>27 °C) as the sun is positioned directly overhead, resulting in maximum incident solar radiation (Xie and Carton, 2004). In this season, the ICTZ is situated close to the equator leading to weak trade winds that cause a deep thermocline in the eastern equatorial Atlantic (EEA). As the year progresses the ITCZ migrates northward, and the southeasterly winds intensify.

This shift leads to a shoaling of the thermocline, enhanced upwelling and vertical mixing as well as intensified evaporation in the EEA (Lübbecke et al., 2018). Consequently, from May to June, the Atlantic cold tongue (ACT) forms east of 20°W, persisting until September with SSTs below 25 °C. The initiation of the ACT and the West African Monsoon (WAM) have



been observed to be interconnected. In fact, delayed onsets of the ACT and WAM are associated with anomalously warm SSTs in the EEA (Brandt et al., 2011; Caniaux et al., 2011).

Every few years, the SST in the EEA experiences large deviations (>1.5 °C) from its climatology, exerting significant influence on the climate of the neighbouring continents (Hirst and Hastenrath, 1983; Folland et al., 1986; Nobre and Shukla, 1996). The EEA depicts notable interannual SST variability, particularly in boreal summer, within the ATL3 region (Zebiak (1993); 20°W-0°E, 3°S-3°N, indicated by the blue box in Figure 2). The enhanced interannual SST variability in May-June-July (MJJ), during the development of the ACT, is the result of the Atlantic zonal mode or Atlantic Niño mode (Servain et al.,

1982; Zebiak, 1993; Keenlyside and Latif, 2007; Lübbecke et al., 2018). Atlantic Niños (Niñas) are characterised by warm (cold) SST anomalies developing in the ATL3 region. The underlying dynamics of the Atlantic Niño bear some resemblance to that observed during El Niño/Southern Oscillation (ENSO) in the Pacific ocean (Zebiak, 1993). It involves a coupling between SST, zonal wind stress and ocean heat content as described by the Bjerknes feedback (Bjerknes, 1969). The Bjerknes feedback can be decomposed into three components: (1) the forcing of western equatorial Atlantic (ATL4; 40°W-20°W, 3°S-3°N; green

box in Figure 2a) zonal wind anomalies by SST anomalies in the ATL3 region; (2) the forcing of thermocline depth anomalies in the ATL3 region by zonal wind anomalies in the ATL4; (3) the forcing of SST anomalies in the ATL3 by local thermocline depth anomalies. All three components of the Bjerknes feedback are active in the equatorial Atlantic although they are generally weaker than those observed in the Pacific (Keenlyside and Latif, 2007; Lübbecke and McPhaden, 2017; Dippe et al., 2019).

Despite substantial warm biases found in state-of-the-art coupled general circulation models (CGCM) in the EEA (Davey

et al., 2002; Richter and Tokinaga, 2020; Farneti et al., 2022), CGCMs are still capable of reproducing the Bjerknes feedback (Deppenmeier et al., 2016). A number of them manage to simulate realistic interannual SST variability within the ATL3 during boreal summer (Figure 2b; see also Richter and Tokinaga, 2020)). However, the CGCM ensemble mean depicts too weak interannual SST variability in the EEA during boreal summer and excessive variability off the coasts of Angola and Namibia (Figure 2b). CGCMs have been extensively evaluated in the tropical Atlantic region, serving as valuable tools for

comprehending and predicting variability patterns (Crespo et al., 2022; Prigent et al., 2023a, b). To our knowledge, relatively little effort has been devoted to the simulation of interannual variability in OGCMs in the tropical Atlantic. For example, Wen et al. (2017) analysed the response of tropical ocean simulations with two different surfaces forcings: NCEP/DOE-R2 (Kanamitsu et al., 2002) and CFSR (Saha et al., 2010). They found that the ocean temperature variability simulated using these two surface forcings was comparable in the tropical Pacific, however, they showed that using CFSR lead to some improvements

in the tropical Atlantic, indicating that the improvements in the tropical Atlantic were mainly attributable to differences in surface winds.

The Ocean Model Intercomparison Project (OMIP; Griffies et al., 2016) provides an ideal framework for evaluating the simulation of interannual SST variability in the equatorial Atlantic by ocean models. The main objective of OMIPs is to provide a framework for assessing, understanding and improving the ocean and sea-ice components of global climate models

that contribute to the Coupled Model Intercomparison Project (CMIP). OMIP have used two atmospheric and river runoff datasets to force ocean sea-ice models. In OMIP phase 1 (OMIP1; Griffies et al., 2009), the Coordinated Ocean-ice Reference Experiments phase-II atmospheric state (CORE-II; Large and Yeager, 2009), mainly derived from the National Centers for





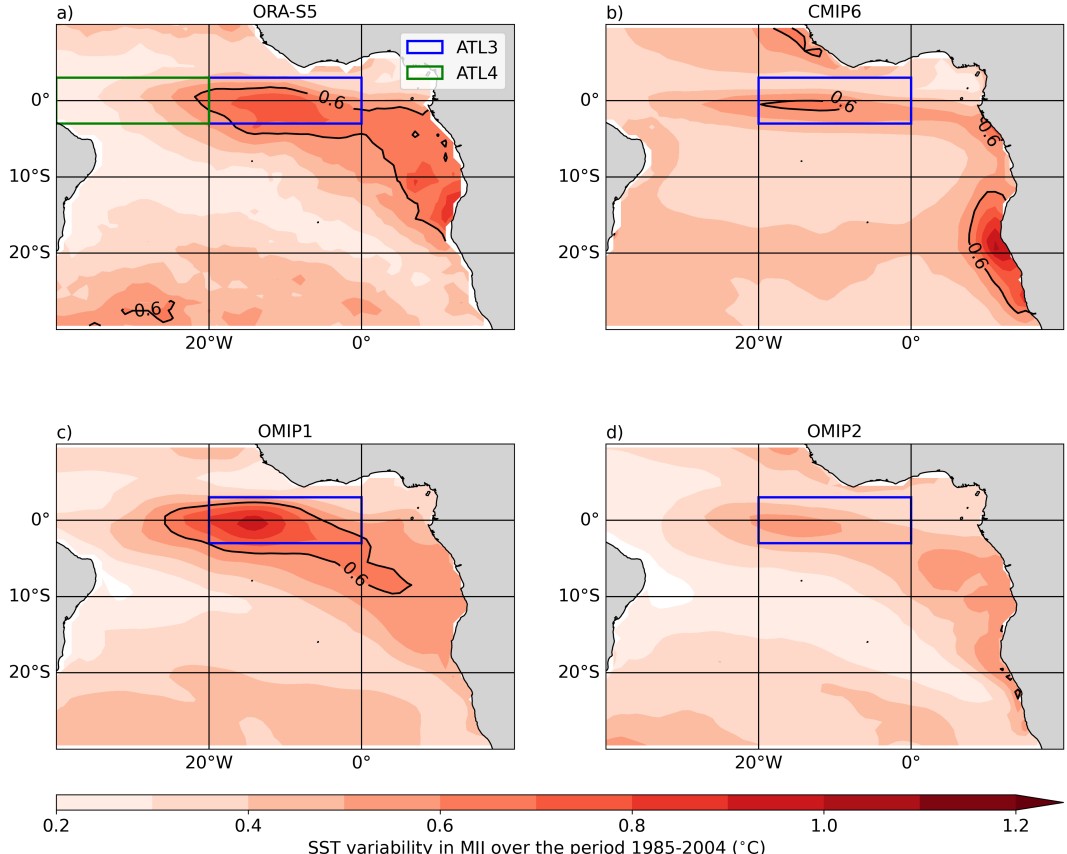

**Figure 2.** Interannual SST variability in the tropical Atlantic during MJJ. Standard deviation of the MJJ-averaged SST anomalies for (a) ORA-S5, (b) CMIP6 ensemble mean, (c) the OMIP1 ensemble mean, and (d) the OMIP2 ensemble mean spanning from January 1985 to December 2004. The blue and green boxes represent the ATL3 (20°W-0°E, 3°S-3°N) and ATL4 (40°W-20°W, 3°S-3°N) regions, respectively.

Environmental Prediction (NCEP) atmospheric reanalysis, was employed. In OMIP phase 2 (OMIP2; Griffies et al., 2016; Tsujino et al., 2020), the JRA-55 based surface dataset for driving ocean–sea-ice models (JRA-55-do; Tsujino et al., 2018) was used.


Compared to ORA-S5 (Figure 2a) over the period January 1985 to December 2004, the OMIP1 ensemble mean overestimates the interannual SST variability in the equatorial Atlantic during boreal summer (Figure 2c), while the OMIP2 ensemble mean underestimates it (Figure 2d). If we extend the time period to the period from January 1970 to December 2004 (Figure S1), relative to ORA-S5, the OMIP1 (OMIP2) ensemble mean still overestimates (underestimates) the MJJ interannual SST variability. When compared to the OMIP1 individual ensemble members over the period from January 1985 to December 2004, the interannual SST variability in the tropical Atlantic during boreal summer is systematically weaker in the OMIP2 ensemble members (Figure S2), raising several questions that we address in the following sections. In particular, how well do OMIP1






and OMIP2 ensembles simulate the seasonal cycles of equatorial Atlantic zonal winds, sea level anomalies (SLA) and SSTs? What are the differences in the interannual SST variability within the EEA between OMIP1 and OMIP2? Is the interannual

SST variability in the equatorial Atlantic, as simulated by OMIP models, dependent on the atmospheric forcing used? If so, which component of the atmospheric forcing leads to discrepancies in the interannual SST variability?

To address these questions we utilise various observational datasets, reanalysis products, and conduct sensitivity experiments, all of which are detailed in section 2. We scrutinise the seasonal patterns of equatorial Atlantic zonal winds, SLAs and SSTs in section 3. Section 4 is dedicated to assessing the interannual SST and temperature variability within OMIP1 and OMIP2

ensembles. In section 5, we delve into the impact of wind forcing on interannual SST variability in the EEA. Final conclusions along with discussions can be found in section 6.

## 2 Data and methods

### 2.1 Data

#### 2.1.1 Reanalysis products and observational datasets

This study employs several reanalysis products with monthly temporal resolution, if not stated otherwise. Specifically, SST, sea surface height (SSH), zonal wind stress, and upper 200 m depth ocean temperature are taken from the Ocean Reanalysis System version 5 (ORA-S5; Zuo et al., 2019). ORA-S5 provides data at a horizontal resolution of $0.25°$ by $0.25°$ and spans the period from January 1958 to present day. ORA-S5 has 72 z-levels in the ocean. The Optimum Interpolation SST version 2 (OI-SST, Reynolds et al., 2002) is also used, it is available at a horizontal resolution of $1°$ by $1°$ over the period from December 1981

to January 2023. Additionally, zonal wind at 10 m height (U10) is obtained from various sources, including the NCEP/NCAR Reanalysis 1 (NCEP-R1; Kalnay et al., 1996), which is available at a horizontal resolution of $2.5°$ by $2.5°$ and covers the period from January 1948 to present day, the NCEP/DOE Reanalysis 2 (NCEP/DOE-R2; Kanamitsu et al., 2002), which is available at a horizontal resolution of $2.5°$ by $2.5°$ from January 1979 to present day, the fifth generation of the European Centre for Medium-Range Weather Forecast (ECMWF) reanalysis (ERA5; Hersbach et al., 2023), with a horizontal resolution of $0.25°$

by $0.25°$, spanning the period January 1940 to present day, the CORE-II forcing (Large and Yeager, 2009), with a horizontal resolution of $2°$ by $2°$ and a temporal resolution of 6 hours encompassing the period from January 1948 to December 2009, and finally the JRA55-do forcing derived from the Japanese 55 years Reanalysis (Griffies et al., 2016; Tsujino et al., 2018), with a horizontal resolution of $0.5625° \times 0.5625°$ ($\sim$ 55 km $\times$ 55 km) and a temporal resolution of 3 hours spanning from January 1958 to December 2018.

In addition to reanalysis products, one zonal wind at 10 m height dataset is obtained from the Cross-Calibrated Multi-Platform version 2 (CCMP v2; Mears et al., 2019), providing data at a horizontal resolution of $0.25°$ by $0.25°$ and spanning from January 1987 to December 2015. To validate the SLA from the OMIP models, we compare it to the monthly mean gridded AVISO data version vDT2021 available at a horizontal resolution of $0.25°$ by $0.25°$ spanning the period from January 1993 to present.



**Table 1.** OMIP1 models (0-5) and OMIP2 models (6-12) used in this study.

| Num | Model name | Ocean model | Ocean resolution (lon × lat × levels) |
|---|---|---|---|
| 0 | CMCC-CM2-SR5 | NEMO3.6 | 362 × 292 × 50 |
| 1 | CMCC-ESM2 | NEMO3.6 | 362 × 292 × 50 |
| 2 | EC-Earth3 | NEMO3.6 | 362 × 292 × 75 |
| 3 | IPSL-CM6A-LR | NEMO-OPA | 362 × 332 × 75 |
| 4 | MRI-ESM2-0 | MRI.COM4.4 | 360 × 364 × 61 |
| 5 | NorESM2-LM | MICOM | 360 × 384 × 70 |
| 6 | ACCESS-OM2 | MOM5.1 | 360 × 300 × 50 |
| 7 | ACCESS-OM2-025 | MOM5.1 | 1440 × 1080 × 50 |
| 8 | CMCC-CM2-HR4 | NEMO3.6 | 1442 × 1051 × 50 |
| 9 | CMCC-CM2-SR5 | NEMO3.6 | 362 × 292 × 50 |
| 10 | EC-Earth3 | NEMO3.6 | 362 × 292 × 75 |
| 11 | MRI-ESM2-0 | MRI.COM4.4 | 360 × 364 × 61 |
| 12 | NorESM2-LM | MICOM | 360 × 384 × 70 |

### 2.1.2 OMIP data

In this study, we assess how models participating in OMIP1 and OMIP2 simulate interannual SST variability in the tropical Atlantic. The OMIP1 protocol consists of five consecutive cycles of the 62-year-long CORE-II forcing (Large and Yeager, 2009), whereas the OMIP2 protocol consists of six consecutive cycles of the 60-year-long JRA55-do forcing. The JRA55-do has a higher temporal resolution (3-hours) and finer spatial resolution ($0.5625° × 0.5625° \sim 55$ km × 55 km) than the CORE-II forcing (6 hourly and 2° by 2°, respectively). For the purpose of analysis, we focused on the fifth and sixth cycle of OMIP1 and OMIP2, respectively, during a common period from January 1985 to December 2004, aligning with Farneti et al. (2022). Only ocean models with a resolution finer than 1° by 1° are considered in this study and are listed in Table 1. All models were bi-linearly interpolated horizontally onto a regular 1° by 1° grid and vertically on the following depth levels: 6, 15, 25, 35, 45, 55, 65, 75, 85, 95, 105, 115, 125, 135, 145, 156.9, 178.4, 222.5, 303.1. The following variables were utilised in the analysis: SST (variable name: TOS), SSH (variable name: ZOS), zonal wind stress (variable name: UAS), ocean temperature (variable name: THETAO), mixed layer depth (variable name: MLOTST), net surface heat flux (variable name: HFDS), and ocean vertical velocity (variable name: WO).

### 2.1.3 CMIP6 data

The eighteen CMIP6 models considered in Figure 2 are listed in Table S1. We use monthly mean outputs of SST from the variant r1i1p1f1 retrieved over the historical period from January 1985 to December 2004. Before analysis, models' outputs were bi-linearly interpolated on a common 1° by 1° grid.





### 2.1.4   Simulations with the GFDL-MOM5 model

We conducted several experiments to complement the OMIP analyses. We employed the NOAA-GFDL Modular Ocean Model version 5 (MOM5; Griffies, 2012), which is a free-surface primitive equation model and uses a z⋆-vertical coordinate.

First, we performed a control run (MOM5-LR) following the OMIP2 protocol (Griffies et al., 2016), running the MOM5 ocean model for six consecutive cycles of the 60-year-long JRA55-do forcing. The simulation was conducted at 1° nominal resolution in the horizontal and 50 vertical levels. In MOM5-LR, subgrid mesoscale processes are parameterized with the Gent-McWilliams skew-flux closure scheme (Gent and Mcwilliams, 1990; Gent et al., 1995; Griffies, 1998) and submesoscale eddy fluxes according to Fox-Kemper et al. (2008) and Fox-Kemper et al. (2011). Vertical mixing is represented with a K-profile parameterization (Large et al., 1994).

Next, we examined the relative influence of different wind and heat flux forcing on interannual SST variability in the EEA by designing a suite of sensitivity experiments. The role of surface winds was investigated with MOM5-LR-winds. This experiment mirrors the MOM5 configuration, with the exception that we repeated the sixth cycle by replacing the JRA55-do winds at 10 m height (U10 and V10) with the CORE-II wind datasets (Large and Yeager, 2009). Although not energetically consistent, this configuration will provide a sensitivity for the upper ocean response to wind forcing. For an investigation into the impact of heat flux forcing on EEA interannual SST variability, we conducted a similar experiment with MOM5-LR-heat. MOM5-LR-heat is similar to MOM5-LR-winds, but in this case the JRA55-do longwave and shortwave heat fluxes are replaced by the CORE-II heat fluxes over the full sixth cycle. It is worth noting that the latent and sensible heat fluxes in MOM5-LR-heat depend on JRA55-do winds. Finally, to assess the impact of the horizontal resolution on the simulation of interannual SST variability in the EEA, we conducted a MOM5-HR experiment following the OMIP2 protocol. MOM5-HR has a similar configuration to MOM5-LR, but its horizontal resolution is refined to 0.25° by 0.25° and the parameterization for mesoscale eddy fluxes is turned off. As for OMIP2 and MOM5-LR, we analysed the sixth cycle of MOM5-HR.

## 2.2   Methodology

### 2.2.1   Definition of anomalies

We compare the interannual SST variability in the EEA simulated by the OMIP1 ensemble mean to the OMIP2 ensemble mean over a 20-year period spanning from January 1985 to December 2004. Throughout this paper, prior to all analysis, the linear trend is removed pointwise to each dataset. Monthly-mean anomalies are computed by subtracting the climatological monthly-mean seasonal cycle derived over the study period. The boreal summer interannual variability is quantified as the standard deviation of the MJJ-averaged anomalies.

### 2.2.2   Thermocline depth, mixed layer depth and sea level anomaly definitions

The mean depth of the thermocline is defined as the depth of the maximum vertical temperature gradient (dT/dz). SSH anomalies are used as a proxy for thermocline-depth variations. Mixed layer depth (MLD) is determined as the ocean depth at which




the density $\sigma_\theta$ has increased by 0.03 kg·m$^{-3}$ relative to the top model level value (Griffies et al., 2016). A discussion on the method and its implications in defining MLD in the OMIP models can be found in Treguier et al. (2023). The MLD, dT/dz

and corresponding depth of the maximum dT/dz are depicted for each OMIP model and sensitivity experiments in Figure S3. Sea level anomaly (SLA) is defined as the pointwise difference between the SSH and the mean sea surface, with the mean sea surface calculated as the SSH averaged of the period January 1985 to December 2004. The equatorial Atlantic thermocline tilt is defined as the difference of the depth of the maximum dT/dz between the ATL4 and ATL3 regions.

### 2.2.3   Bjerknes feedback and thermal damping

The three components of the Bjerknes feedback (BF) are assessed as follows. The first component (BF1) is the linear regression of ATL4-averaged zonal wind stress anomalies in MJJ on ATL3-averaged SST anomalies in MJJ. The second component (BF2) is the linear regression of ATL3-averaged SSH anomalies in MJJ on ATL4-averaged zonal wind stress anomalies in MJJ. The third component (BF3) is the linear regression of ATL3-averaged SST anomalies in MJJ on ATL3-averaged SSH anomalies in MJJ. Additionally, the thermal damping is quantified as the linear regression of ATL3-averaged net heat flux anomalies in MJJ

on ATL3-averaged SST anomalies in MJJ.

### 3   Comparison of the OMIP1 and OMIP2 seasonal cycles

In this section we compare the OMIP1 and OMIP2 ensemble mean seasonal cycles of the equatorial Atlantic (40°W-10°E; 3°S-3°S) zonal winds, SLA and SST to the reanalysis products and observational datasets.

The seasonal cycle of the zonal wind in the western equatorial Atlantic is dominated by an annual cycle with maximum

easterly winds in September-October-November (SON) and minimum easterlies in MAM (Figure 3a). Meanwhile, the EEA zonal wind exhibits a semiannual cycle (Figure 3a) with maxima in Januray-February-March and in SON. Both CORE-II (Figure 3b) and JRA55-do (Figure 3c) surface forcings closely mirror the observed seasonal cycles of the zonal wind in the equatorial Atlantic.

Next, we analyse the seasonal cycle of the SLA, where negative (positive) SLA indicates a shoaling (deepening) of the

thermocline. Consistent with the strong link between zonal winds and the thermocline depth (Philander and Pacanowski, 1986), the seasonal cycle of the equatorial Atlantic thermocline depicts an annual cycle in the west and a semiannual cycle in the east (Figure 3d; Ding et al., 2009). In the western equatorial Atlantic, the thermocline reaches its shallowest point during MAM and this signal progresses eastward, reaching 10°W by July. In SON, the thermocline is deep in the west and the signal also propagates eastward, but its propagation is faster. These eastward propagating SLA signals can be understood in terms of

linear dynamics and are essentially explained by the first four baroclinic modes (Ding et al., 2009). In the EEA, a semiannual SLA cycle emerges. Both the OMIP1 (Figure 3e) and OMIP2 (Figure 3f) ensemble means exhibit patterns similar to ORA-S5. However, the amplitude of the annual cycle in the western equatorial Atlantic is weaker in the OMIP2 ensemble mean compared to the OMIP1 ensemble mean. Quantitatively, the difference in SLA between SON and MAM, averaged between 40°W and 30°W and from 3°S to 3°N, amounts to 0.096 m, 0.072 ± 0.007 m and 0.053 ± 0.01 m for ORA-S5, OMIP1,



and OMIP2 ensemble means, respectively. Hence, relative to the OMIP2 ensemble mean, the annual cycle of the SLA in the western equatorial Atlantic is 35% larger in the OMIP1 ensemble mean. Yet, relative to ORA-S5, both OMIP ensemble means have a too weak SLA annual cycle in the western equatorial Atlantic.

The shoaling of the thermocline depth from MAM to JAS in the ATL3 region is closely related to the rapid decrease in SST (Figure 3g). For ORA-S5, the ATL3-averaged SST drops by 3.73 °C, decreasing from 28.52 °C in MAM to 24.79 °C in JAS. In the case of the OMIP1 ensemble (Figure 3h), the ATL3-averaged SST drops by $3.18 \pm 0.1$ °C, declining from $28.53 \pm 0.05$ °C in MAM to $25.35 \pm 0.08$ °C in JAS. Similarly, for the OMIP2 ensemble (Figure 3i), the ATL3-averaged SST decreases by $3.28 \pm 0.16$ °C, going from $28.65 \pm 0.05$ °C in MAM to $25.37 \pm 0.16$ °C in JAS. These numbers indicate that, in comparison to ORA-S5, both OMIP ensemble means generate a weaker cooling from MAM to JAS. It is noteworthy that the difference in cooling between the OMIP1 and OMIP2 ensemble means is due to slightly warmer ATL3 SSTs in MAM for OMIP2 compared to OMIP1.

As the seasonal cycle of the equatorial Atlantic SST is strongly influenced by subsurface conditions, we examined the upper 200 m ocean temperature during both MAM and JAS (Figure 4). During MAM (Figure 4a), the easterly winds in the equatorial Atlantic (40°W-10°E; 3°S-3°N) are relatively weak, measuring 1.91 m·s$^{-1}$ in CCMP v2. Consequently, the thermocline exhibits a small tilt of 23.30 m, with the upper 25 m in the ATL3 region having a temperature of 28.44 °C. Notably, the ATL3-averaged MLD is located at 18.87 m. We note that the vertical temperature gradient is pronounced in this region, with the distance between the 20 °C and 25 °C isotherms measuring 19.68 m. In JAS (Figure 4b), the equatorial Atlantic easterlies intensify to 2.24 m·s$^{-1}$, leading to a steeper thermocline with a of tilt 44.45 m and an increased slope of the isotherms between 20°W and 0°E. The upper 25 m in the ATL3 experiences a strong cooling, with a temperature of 24.67 °C, while the MLD deepens to 26.26 m.

Comparing the above values to the OMIP1 ensemble mean, we observe that the upper 200 m temperature sections in both MAM (Figures 4c and d) and JAS (Figures 4e and f) align closely with ORA-S5. However, some differences are listed next. In MAM (Figure 4c), the CORE-II easterlies in the equatorial Atlantic remain weak at 1.89 m·s$^{-1}$, resulting in a thermocline tilt of $30.36 \pm 3.79$ m. The upper 25 m in the ATL3 region exhibit a temperature of $28.40 \pm 0.05$ °C, and the maximum vertical velocity, found at 35 m depth, amounts to $4.17 \pm 0.35$ $10^{-6}$ m·s$^{-1}$ (Figure 4d). The ATL3-averaged MLD is located at $16.36 \pm 1.53$ m. Notably, the distance between the 20 °C and 25 °C isotherms in the ATL3 region is larger (40.61 m) in the OMIP1 ensemble, indicating a more diffusive thermocline. In JAS (Figure 4e), the CORE-II easterlies strengthen to 2.76 m·s$^{-1}$, resulting in a thermocline tilt of $46.64 \pm 3.36$ m. The upper 25 m temperature in the ATL3 decreases to $25.26 \pm 0.11$ °C, and the maximum vertical velocity deepens to 55 m depth and strengthens to $5.64 \pm 1.12$ $10^{-6}$ m·s$^{-1}$ (Figure 4f). The ATL3-averaged MLD deepens to $23.91 \pm 2.92$ m.

We next focus on the OMIP2 ensemble mean (Figures 4g-j). In MAM (Figures 4g-h), the JRA55-do easterlies in the equatorial Atlantic, measuring 1.99 m·s$^{-1}$, are stronger than the CORE-II ones, leading to a larger thermocline tilt of $35.04 \pm 3.72$ m. The ATL3-averaged upper 25 m temperature reaches to $28.44 \pm 0.12$ °C, and the MLD is located at $13.90 \pm 3.44$ m. More importantly, the distance between the 20 °C and the 25 °C remains substantial at 41.13 m. The maximum vertical velocity in the ATL3 (Figure 4h), situated at 35 m depth, is $4.24 \pm 0.74$ $10^{-6}$ m·s$^{-1}$. In JAS (Figure 4f), the JRA55-do easterlies





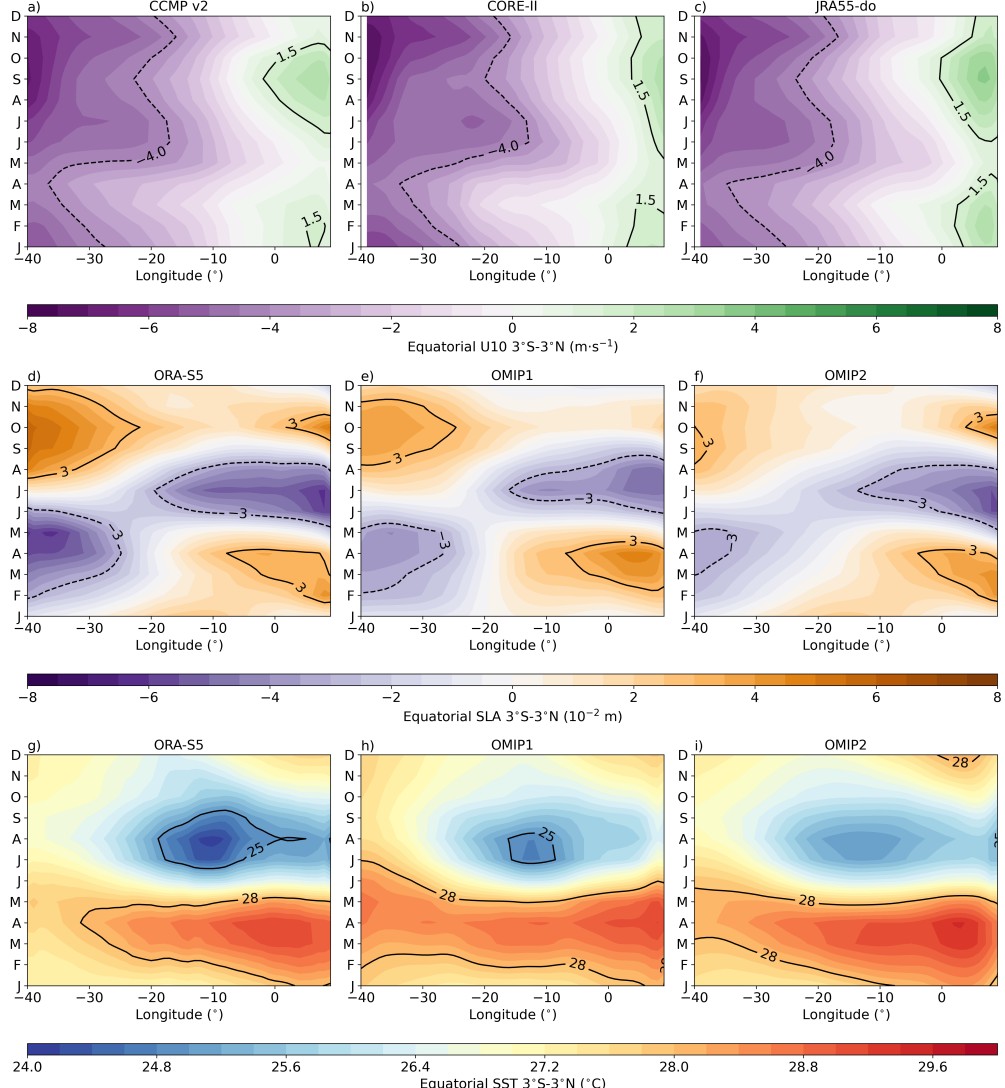

**Figure 3.** Hovmöller diagrams illustrating seasonal cycles in the equatorial Atlantic U10, SLA and SST. (a) The seasonal cycle of CCMP v2 U10, averaged between 3°S and 3°N and presented as a function of longitude and calendar month for the period January 1987 to December 2004. (b) The seasonal cycle of CORE-II zonal wind, averaged between 3°S and 3°N, displayed as a function of the longitude and calendar month for the period Janurary 1985 to December 2004. (c) Same as (b) but for the JRA55-do U10. (d) The seasonal cycle of SLA in ORA-S5, averaged between 3°S and 3°N, shown as a function of the longitude and calendar month for the period from January 1985 to December 2004. (e) Similar to (d) but for the OMIP1 ensemble mean. (f) Same as (d) but for the OMIP2 ensemble mean. (g) the seaonal cycle of SST in ORA-S5, averaged between 3°S and 3°N, presented as a function of longitude and calendar month for the period January 1985 to December 2004. (h) Similar to (g) but representing the OMIP1 ensemble mean. (i) Same as (g) but for the OMIP2 ensemble mean.





strengthen to 2.27 m·s$^{-1}$, resulting in a thermocline tilt of 44.03 ± 3.59 m. The upper 25 m ATL3 temperature decreases to
25.28 ± 0.19 °C and the maximum vertical velocity deepens to 45 m depth but weakens slightly to 4.19 ± 1.23 $10^{-6}$ m·s$^{-1}$.
The ATL3-averaged MLD deepens to 26.20 ± 5.84 m. We note that the 25 °C isotherm is not tilting as much as in the OMIP1
ensemble mean, consistent with differences in vertical velocities within the ATL3 region (Figures 4h, j).

To summarise this section and answer the first of our questions, we find that the OMIP1 and OMIP2 ensemble means closely
replicate the seasonal cycles of equatorial Atlantic zonal winds, SSTs, SLAs and upper 200 m ocean temperatures when
compared to ORA-S5. Nonetheless, we highlight some discrepancies relative to ORA-S5: (1) the annual cycle of the SLA in
the western equatorial Atlantic is weaker in both OMIP ensemble means, but the OMIP1 ensemble mean annual cycle of the
SLA in the western equatorial Atlantic is 35% larger than the one of the OMIP2 ensemble mean; (2) both OMIP ensemble
means exhibit a too diffusive thermocline; (3) the cooling of the SST from MAM to JAS in the ATL3 is less pronounced in the
OMIP ensemble means; (4) the equatorial upwelling in JAS appears to be weaker in both OMIP ensemble means.

## 4 Comparison of OMIP1 and OMIP2 interannual variabilities

The interannual variability in the equatorial Atlantic exhibits a pronounced seasonality (Keenlyside and Latif, 2007; Lübbecke
et al., 2018). Specifically, high interannual zonal wind variability in CCMP v2 in the western equatorial Atlantic occurs from
40°W to 20°W during April-May-June (Figure 5a) and from 20°W to 15°W in March and April. As OMIP1 and OMIP2
models are forced by the CORE-II and JRA55-do 10 m winds, respectively, we compare them to CCMP v2. The CORE-II
zonal wind forcing displays a similar pattern to CCMP v2 from 40°W to 20°W but with weaker interannual variability (Figure
5b). The JRA55-do forcing also exhibits a similar pattern of interannual zonal wind variability but underestimates it in April-
May-June. Additionally, JRA55-do forcing (Figure 5c) reveals high zonal wind variability between 10°W and 10°E in January
and February, which is not as prominent in CCMP v2 (Figure 5a) and absent in the CORE-II forcing (Figure 5b). Quantitatively,
the standard deviation of AMJ-averaged U10 anomalies in the ATL4 region is 0.80 m·s$^{-1}$ for CCMP v2 over the period from
January 1988 to December 2004 and 0.70 m·s$^{-1}$ and 0.68 m·s$^{-1}$ for CORE-II and JRA55-do over the period from January
1985 to December 2004, respectively.

Typically, sudden relaxation (intensification) of the Trade winds in the western equatorial Atlantic can trigger interannual
downwelling (upwelling) equatorial Kelvin waves (Imbol Koungue et al., 2017). While propagating eastward along the equa-
torial wave guide, these waves generate thermocline-depth variations which can be observed in the SSH anomalies. In the
following we compare the interannual SSH variability in OMIP1 and OMIP2 ensemble means to ORA-S5. In ORA-S5, two
peaks of interannual SSH variability are observed during boreal summer, one between 40°W and 35°W and another between
20°W and 0°E (Figure 5d). Additionally, ORA-S5 exhibits high interannual SSH variability in November-December in the
EEA (Figure 5d). The interannual SSH variability in the ATL3 region is considerably stronger in the OMIP1 ensemble mean
compared to the OMIP2 ensemble mean (Figure 5e, f). In numbers, the OMIP1 (OMIP2) ensemble mean ATL3-averaged SSH
variability in MJJ is 0.02 ± 0.001 m (0.014 ± 0.001 m), while it is 0.019 m in ORA-S5. The anomaly correlation coefficients
and root-mean-square errors between OMIP1 and OMIP2 simulations with AVISO SLA, evaluated over the period January





**Figure 4.** Upper 200 m ocean temperature for MAM (left) and JAS (right). (a, c, g) MAM upper 200 m ocean temperature in the equatorial Atlantic (40°W-9°E, 3°S-3°N) with shading, where black arrows indicate zonal wind at 10 m height, thick blue lines denote the maximum dT/dz, green dashed lines represent the mixed layer depth, and black lines indicate the depths of the 20 °C and 25 °C isotherms for ORA-S5, OMIP1 and OMIP2 ensemble means, respectively. (b, e, i) Same as (a, c, e) but for JAS. (d, h) Vertical ocean velocity profiles averaged over the ATL3 region in MAM for the OMIP1 and OMIP2 ensemble means, respectively. Shaded areas represent the ensembles' spreads, estimated as ± 1 standard deviation. (f, j) Same as (d, h) but for JAS. Vertical black dashed lines denote the ATL3 region.





1993 to December 2004, are shown in Figures S4a-d, respectively. These figures exhibit high correlation (>0.75, Figures S4a, b) and low root-mean-square error (<0.01 m, Figures S4c, d) in the EEA for both OMIP1 and OMIP2 ensemble means,

indicating a good fidelity of the OMIP ensembles with AVISO. To further illustrate that, we show the timeseries depicting ATL3-averaged SSH anomalies for AVISO, OMIP1, and OMIP2 ensemble means in Figure S4e. Despite robust correlations between both OMIP ensembles and AVISO (0.79 for OMIP1 and 0.78 for OMIP2), evaluated over the period from January 1993 to December 2004, the amplitude of the monthly mean SSH anomalies is larger in OMIP1 compared to OMIP2. This indicates that thermocline depth variations are larger in the OMIP1 ensemble mean compared to the OMIP2 ensemble mean.

Finally, we compare the interannual SST variability in the equatorial Atlantic from the OMIP1 and OMIP2 ensemble means to ORA-S5. ORA-S5 displays two peaks of interannual SST variability in the ATL3 region, one in MJJ and one in November-December (Figure 5g). Both OMIP ensemble means exhibit a similar pattern to ORA-S5. However, relative to ORA-S5, the OMIP1 (OMIP2) ensemble mean overestimates (underestimates) the MJJ interannual SST variability (Figure 5h, i). In numbers, the standard deviation of the MJJ-averaged SST anomalies in the ATL3 region is 0.62 $\pm$ 0.04 °C, 0.41 $\pm$ 0.03 °C

and 0.59 °C for the OMIP1 and OMIP2 ensemble means and ORA-S5, respectively. The anomaly correlation coefficients and root-mean-square errors between OMIP1 and OMIP2 simulations with OI-SST, evaluated over the period January 1985 to December 2004, are shown in Figures S5a-d, respectively. In comparison to the tropical Atlantic ocean, the EEA and southeastern tropical Atlantic display the lowest anomaly correlation and greatest root-mean-square errors across both OMIP1 and OMIP2 ensembles. This indicates that these regions exhibit the most pronounce disparities between both OMIP ensembles and OI-SST. Nevertheless, it is important to highlight that despite these differences, the anomaly correlation coefficient is high

($\approx$0.8) and the root mean-square error is low (<0.5 °C). To elaborate on this point we present the timeseries depicting the ATL3-averaged SST anomalies for ORA-S5, OMIP1, and OMIP2 ensemble means in Figure S5e. Both OMIP1 and OMIP2 ensemble means are highly correlated to OI-SST with Pearson correlation coefficients of 0.79 and 0.80, respectively. However, the amplitude of the SST anomalies in the ATL3 region in the OMIP1 ensemble mean are in general larger than in the OMIP2

ensemble mean.

Ocean-atmosphere interactions are key drivers of the interannual SST variability within the EEA (Jouanno et al., 2017). To delve into this, we examined the various components of the Bjerknes feedback and thermal damping in ORA-S5, along with the ensemble means of OMIP1 and OMIP2 (Figure 6) over the period January 1985 to December 2004. In comparison to ORA-S5 (0.91 $10^{-2}$ N·m$^{-2}$·°C$^{-1}$), the BF1 (Figure 6a) is overestimated in both OMIP1 (1.05 $\pm$ 0.05 $10^{-2}$ N·m$^{-2}$·°C$^{-1}$) and OMIP2

(1.23 $\pm$ 0.15 $10^{-2}$ N·m$^{-2}$·°C$^{-1}$). Notably, the BF1 is larger in the OMIP2 ensemble than in the OMIP1 ensemble, but the latter has a narrower spread. Turning to the BF2 (Figure 6b) it amounts to 1.79 m·(N·m$^{-2}$)$^{-1}$ in ORA-S5 and is overestimated (underestimated) in the OMIP1 (OMIP2) ensemble with a slope of 1.88 $\pm$ 0.11 m·(N·m$^{-2}$)$^{-1}$ (1.55 $\pm$ 0.21 m·(N·m$^{-2}$)$^{-1}$). Regarding the BF3 (Figure 6c), it equals to 26.51 °C·m$^{-1}$ for ORA-S5 and to 29.34 $\pm$ 1.8 °C·m$^{-1}$ and 26.10 $\pm$ 1.1 °C·m$^{-1}$ for the OMIP1 and OMIP2 ensembles, respectively. Hence, the subsurface-surface coupling is more pronounced in the OMIP1

ensemble mean than in the OMIP2 ensemble mean. Lastly, the thermal damping is assessed (Figure 6d). While ORA-S5 depicts a strong thermal damping (-21.58 W·m$^{-2}$·°C$^{-1}$) both OMIP ensemble underestimate it, with slopes of -12.18 $\pm$ 1.72 W·m$^{-2}$·°C$^{-1}$ and -10.02 $\pm$ 2.34 W·m$^{-2}$·°C$^{-1}$ for the OMIP1 and OMIP2 ensembles, respectively.



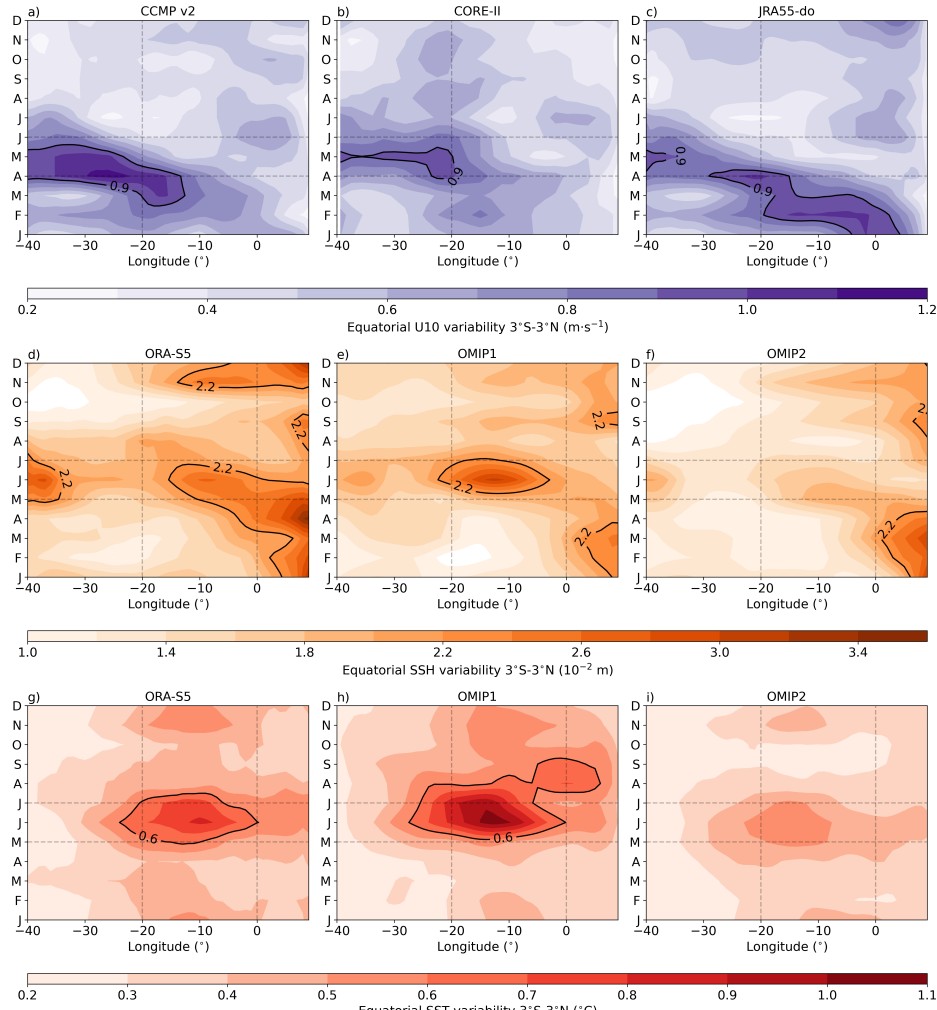

**Figure 5.** Hovmöller diagrams depicting equatorial Atlantic U10, SSH and SST interannual variability. (a) Standard deviation of CCMP v2 U10 anomalies averaged between 3°S and 3°N, plotted as a function of longitude and calendar month, spanning from January 1987 to December 2004. (b) Standard deviation of CORE-II zonal wind anomalies, averaged between 3°S and 3°N, and displayed as a function of longitude and calendar month for the period from January 1985 to December 2004. (c) Same as (b) but for JRA55-do U10. (d) Standard deviation of SSH anomalies in ORA-S5, averaged between 3°S and 3°N, and plotted as a function of longitude and calendar month for the period January 1985 to December 2004. (e) Same as (d) but for the OMIP1 ensemble mean. (f) Same as (d) but for the OMIP2 ensemble mean. (g) Standard deviation of SST anomalies in ORA-S5, averaged between 3°S and 3°N, plotted as a function of longitude and calendar month for the period from January 1985 to December 2004. (h) Same as (g) but for the OMIP1 ensemble mean. (i) Same as (g) but for the OMIP2 ensemble mean.



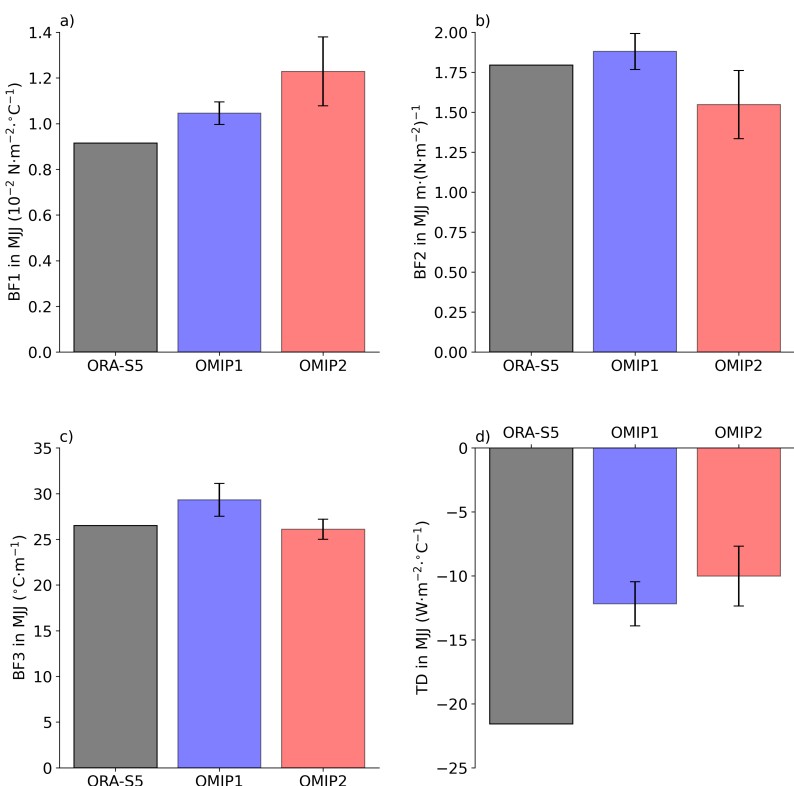

**Figure 6.** Bjerknes feedback components and thermal damping during MJJ over the period from January 1985 to December 2004. (a) Histogram of the BF1 in MJJ for ORA-S5 (grey), the OMIP1 ensemble (blue), and the OMIP2 ensemble (red). (b) Same as (a) but for the BF2. (c) Same as (a) but for the BF3. (d) Same as (a) but for the thermal damping. Error bars are defined as $\pm$ one standard deviation of the ensemble.

The contrast between the interannual variability in the OMIP1 and OMIP2 ensemble means extends beyond the surface, as illustrated by the upper 200 m temperature variability in MJJ (Figure 7). In ORA-S5 (Figure 7a), the maximum interannual temperature variability in MJJ occurs between 40°W and 30°W and from 20°W and 0° within $\pm$ 10 m range around the mean thermocline. The high temperature variability in the western equatorial Atlantic is situated at a depth of 90 m, making it too deep to influence the MLD and, hence, the SST. In contrast, the maximum temperature variability in the EEA is located at 50 m depth, close to the MLD, with an average of 1.28 °C for ORA-S5 when considering the ATL3 region and a $\pm$ 10 m range around the mean thermocline. In MJJ, the interannual temperature variability in the EEA for the OMIP1 ensemble mean (Figure 7b) features a similar pattern to ORA-S5 but with generally weaker temperature variability. The standard deviation of the MJJ-averaged temperature anomalies within $\pm$ 10 m of the mean thermocline for the ATL3 region in the OMIP1 ensemble mean is 0.77 $\pm$ 0.06 °C. The OMIP2 ensemble mean (Figure 7c) also follows a similar pattern to ORA-S5 but displays even less interannual temperature variability in the upper 200 m during MJJ compared to the OMIP1 ensemble mean (Figure 7b).





The standard deviation of the MJJ-averaged temperature anomalies within ± 10 m of the mean thermocline for the ATL3
region in the OMIP2 ensemble mean is 0.58 ± 0.08 °C. The upper 200 m temperature variability in the EEA during MJJ is
systematically larger in the OMIP1 ensemble members compared to the OMIP2 ensemble members as shown in Figure S6.
Given that both OMIP ensemble means exhibit a similar vertical temperature gradient during boreal summer within ± 10 m of
the mean thermocline in the ATL3 region, which is -0.15 ± 0.001 °C·m$^{-1}$, it can be inferred that the disparities in interannual
temperature variability are primarily driven by larger fluctuations in the thermocline depth. In contrast, the boreal summer
vertical temperature gradient for ORA-S5 within ± 10 m of the mean thermocline in the ATL3 region is -0.25 °C·m$^{-1}$, which
can account for its substantially higher subsurface temperature variability.

In response to our second question, related to differences in interannual SST variability in EEA within OMIP1 and OMIP2,
we observed that during the period January 1985 to December 2004 the OMIP1 ensemble exhibits greater boreal summer
interannual SST and temperature variability in the ATL3 region compared to the OMIP2 ensemble. When contrasting the
two ensembles, the OMIP1 ensemble displays a stronger BF2 and BF3, which could account for the larger interannual SST
variability. However, the BF1 component is larger in OMIP2, while the thermal damping is more prominent in the OMIP1
ensemble. In the next section we investigate the impact of the wind forcing on interannual variability in the EEA.

## 5 Influence of the wind forcing on the equatorial Atlantic interannual variability

Wind forcing is an important driver of the equatorial Atlantic mean-state and interannual variability (Richter et al., 2012;
Wahl et al., 2011). Wen et al. (2017) investigated the response of the tropical ocean simulations to NCEP/DOE-R2 and CFSR
surface fluxes. Using sensitivity experiments run with the GFDL MOM version 4p1 (Griffies, 2009), they found that prescribing
CFSR surface fluxes instead of NCEP/DOE-R2 surface fluxes was significantly improving the simulation of the SST and SSH
variabilities in the tropical Atlantic ocean. This underscores the sensitivity of the simulation of the tropical Atlantic interannual
variability to surface forcings. In Figure 8, we compare different wind reanalysis products over the period from January 1985
to December 2004. All of these reanalyses depict a similar seasonal cycle, characterised by the weakest winds in March and
April, followed by a strengthening of the easterly winds from May to December. Notably, NCEP-R1 displays consistently
weaker easterlies throughout the year (Figure 8a). Taking the CCMP v2 climatology as a reference, the root mean square error
of the climatology is 0.39 m·s$^{-1}$ for CORE-II and is 0.27 m·s$^{-1}$ for JRA55-do. When examining zonal wind variability in the
western equatorial Atlantic, the amplitude of zonal wind variability between CORE-II and JRA55-do is relatively similar, but
JRA55-do reaches its peak in April, while CORE-II's peak occurs in May (Figure 8b). The examination of extended timeseries
of zonal wind anomalies in the ATL4 region for both the JRA55-do (over the period 1958 to 2022) and the CORE-II (over
the period 1948 to 2007), shows that both forcing datasets of their maximum variability in May (Figure S7). This highlights
that the disparity in the timing of the peak of the zonal wind variability between JRA55-do and CORE-II (Figure 8b) within
the ATL4 region is linked to the specific time periods under consideration. ERA5 depicts the largest zonal wind variability
and peaks in April. Both NCEP/DOE-R2 and NCEP-R1 exhibit the smallest zonal wind variability, with both reaching their
peaks in May. Taking the CCMP v2 seasonal cycle of the zonal wind variability as a reference, the root-mean-square error for



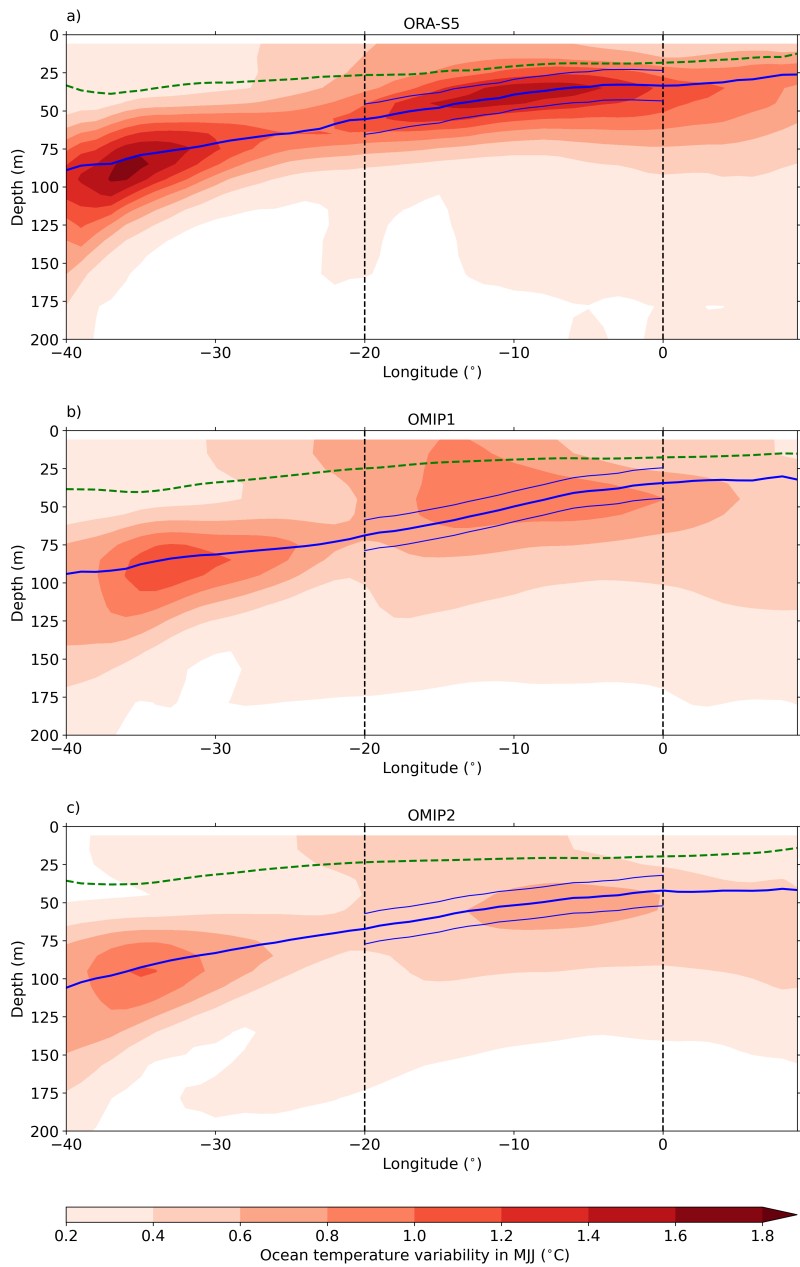

**Figure 7.** Interannual variability of upper 200 m ocean temperature during MJJ over the period from January 1985 to December 2004. Upper 200 m ocean temperature interannual variability in the equatorial Atlantic (40°W-9°E, 3°S-3°N) for (a) ORA-S5, (b) the OMIP1 ensemble mean, and (c) the OMIP2 ensemble mean. Thick blue lines represent the maximum dT/dz, while green dashed lines denote the mixed layer depth. Thin blue lines encompass a ± 10 m range around the mean thermocline. Vertical dashed lines in black denote the ATL3 region.





CORE-II is 0.15 m·s$^{-1}$ and it is 0.09 m·s$^{-1}$ for JRA55-do. The correlation matrix, spanning January 1985 to December 2004, shows the Pearson correlation coefficients between the ATL4-averaged zonal wind anomalies from the different reanalysis datasets (Figure S8). Specifically, it shows that the Pearson correlation coefficient between CCMP v2 and CORE-II is 0.81,
while it is 0.89 for JRA55-do (Figure S8). In addition, the Pearson correlation coefficients for ERA5 in relation to CORE-II and JRA55-do are 0.82 and 0.90, respectively (Figure S8). Additionally, we note that JRA55-do, ERA5 and NCEP/DOE-R2 have a secondary peak in February (Figure 8b), whereas this secondary peak is notably weaker in CORE-II and NCEP-R1. In comparison to the CORE-II forcing, the peak of zonal wind variability during February found in the JRA55-do forcing results from a few strong events occurring between 1985 and 2004 (Figure S9a). Figures S9b and S9c depict zonal wind anomalies in
the western equatorial Atlantic for April and May, respectively. These figures highlight that zonal winds anomalies in April are more pronounced in JRA55-do compared to CORE-II, while the reverse is observed in May.

Consequently, in the following, we aim to examine the hypothesis that the different interannual SST variabilities observed in the OMIP ensemble means are a direct consequence of the discrepancies in wind forcing. To investigate this, we employ two additional simulations, namely MOM5-LR and MOM5-LR-winds, as described in section 2.1.4 and compared in Figure 9.

Both MOM5-LR (Figure 9a) and MOM5-LR-winds (Figure 9b) depict high boreal summer interannual SST variability within the ATL3 region. Yet, the MOM5-LR-winds simulation exhibits a larger interannual SST variability, amounting to 0.59 °C, in contrast to 0.42 °C for MOM5-LR. This implies that solely replacing JRA55-do winds with CORE-II winds results in a 40% increase in interannual SST variability in the EEA. Furthermore, this increase is not limited to the surface, as it is also reflected in the upper 200 m temperature variability during boreal summer. Specifically, the interannual temperature variability
within a ± 10 m range around the mean thermocline is 0.49 °C for MOM5-LR (Figure 9c) and 0.71 °C for MOM5-LR-winds (Figure 9d). Hence, using CORE-II instead of JRA55-do winds leads to a 45% increase in boreal summer temperature variability in the ATL3 region and within ±10 m of the mean thermocline.

The impact of the wind forcing on interannual SSH variability is examined in Figure 10. In comparison to MOM5-LR (Figure 10a), MOM5-LR-winds (Figure 10b) exhibits a similar pattern of equatorial Atlantic interannual SSH variability, albeit with
a larger magnitude. Quantitatively, the standard deviation of MJJ-averaged SSH anomalies in the ATL3 region amounts to 0.015 m for MOM5-LR and 0.017 m for MOM5-LR-winds. Furthermore, there is a shift of one month from June to July of the maximum interannual SSH variability in the EEA. This temporal shift of one month in interannual SSH variability could be related to the different peaks in zonal wind variability in the ATL4 region for JRA55-do (April) and CORE-II (May). As discussed previously, we also find increased SST variability in MOM5-LR-winds (Figure 10d) relative to MOM5-LR (Figure
10c). As for the ATL3 interannual SSH variability, a shift of one month from June to July is also observed in the interannual SST variability in MOM5-LR-winds.

This section allows to answer the last question raised in the introduction. We have demonstrated that the surface forcing, and in particular the wind forcing, has a significant impact on the interannual SST variability in the equatorial Atlantic. Indeed, substituting the JRA55-do winds by the CORE-II winds in MOM5-LR results in a substantial 40% increase in ATL3 interannual
SST variability during MJJ, rising from 0.42 °C for MOM5-LR to 0.59 °C for MOM5-LR-winds.




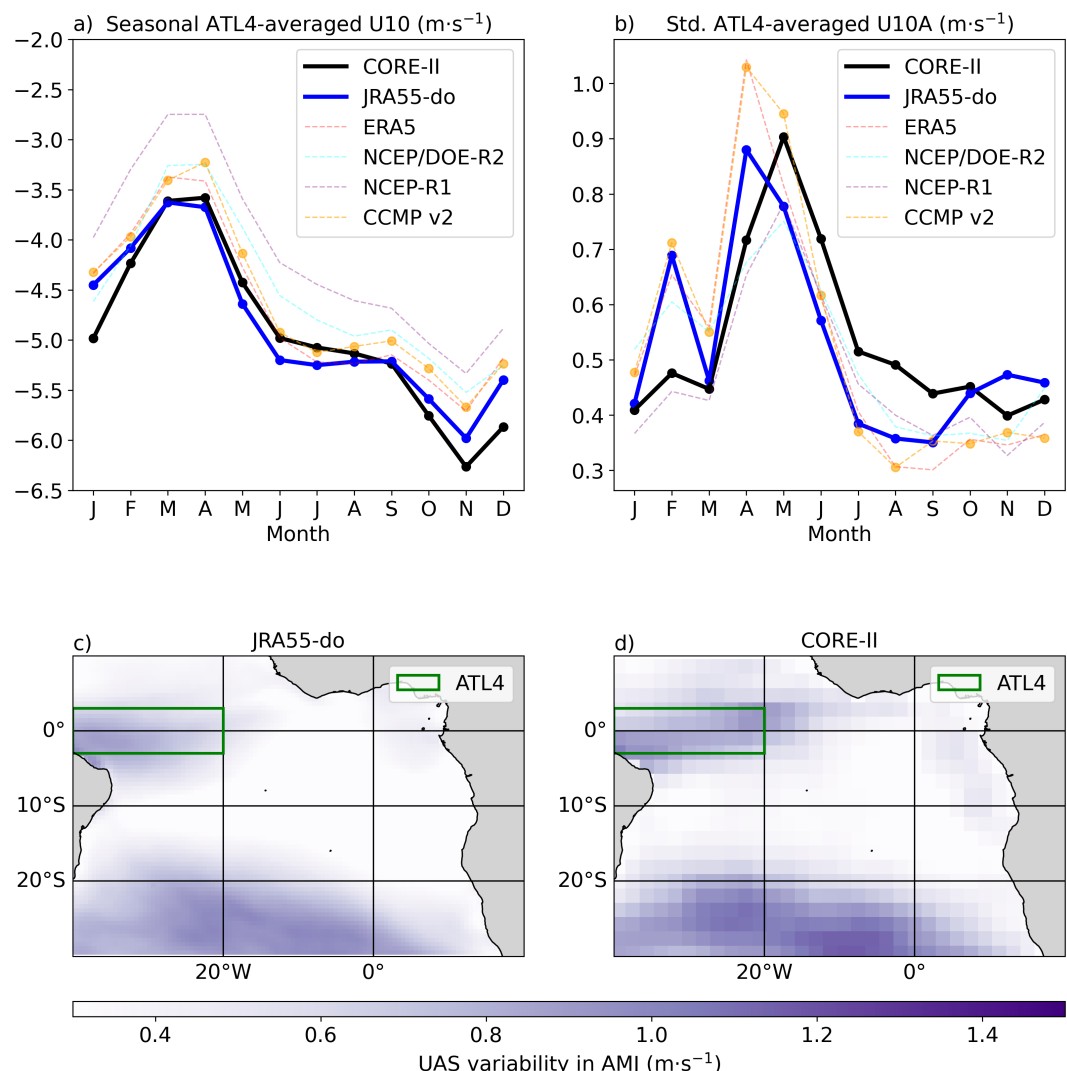

**Figure 8.** Western equatorial Atlantic U10 seasonal cycles for the period from January 1985 to December 2004. (a) Seasonal cycle of U10 winds averaged over the ATL4 region. (b) Seasonal cycle of the standard deviation of U10 anomalies averaged over the ATL4 region. Different lines correspond to various reanalysis products: (black) CORE-II, (blue) JRA55-do, (red) ERA5, (cyan) NCEP/DOE-R2, (purple) NCEP-R1, and (orange) CCMP v2. (c, d) Standard deviation of U10 anomalies over the tropical Atlantic during AMJ for JRA55-do and CORE-II respectively.



**Figure 9.** Interannual SST and temperature variability during MJJ over the period from January 1985 to December 2004. Standard deviation of SST anomalies averaged over MJJ for (a) MOM5-LR and (b) MOM5-LR-winds. Standard deviation of temperature anomalies within the upper 200 m depth of the equatorial Atlantic (3°S-3°N) during MJJ for (a) MOM5-LR and (b) MOM5-LR-winds. The dashed green lines represent the MLD. The solid blue lines indicate the depth of the maximum vertical temperature gradient in MJJ. Thin blue lines encompass a ± 10 m range around the mean thermocline. Vertical dashed black lines denote the ATL3 region.

# 6  Conclusions and discussions

## 6.1  Conclusions

In this study, we have compared the seasonal cycle of equatorial Atlantic zonal wind, SLA, and SST between the OMIP1 and OMIP2 ensemble means and ORA-S5. Furthermore, we conducted an investigation into the interannual variability of the EEA



**Figure 10.** Hovmöller diagrams depicting the interannual variability of SSH and SST over the period from January 1985 to December 2004. (a) Standard deviation of the MOM5-LR SSH anomalies, averaged between 3°S and 3°N, plotted as a function of longitude and calendar month. (b) Same as (a) but for MOM5-LR-winds. (c) Standard deviation of the MOM5-LR SST anomalies, averaged between 3°S and 3°N, plotted as a function of longitude and calendar month. (d) Same as (c) but for MOM5-LR-winds.



within the OMIP models. Finally, we delved into the causes behind the distinct interannual SST variability in OMIP1 and OMIP2 ensembles using sensitivity experiments. We have demonstrated that over the period from January 1985 to December 2004:

- The seasonal patterns of the equatorial Atlantic zonal wind, SLA, SST, and ocean temperature in OMIP1 and OMIP2 ensemble means resemble those in ORA-S5. However, some discrepancies are evident: the annual cycle of the SLA in the western equatorial Atlantic is weaker in both OMIP ensemble means, but the OMIP1 ensemble mean annual cycle of the SLA in the western equatorial Atlantic is 35% larger than the one of the OMIP2 ensemble mean; both OMIP ensembles have a too diffuse thermocline; the cooling of SST from MAM to JAS is insufficient in both OMIP ensembles (Figure 3); and the OMIP2 ensemble mean maximum vertical ocean velocity in JAS is weaker than in the OMIP1 ensemble mean. Regarding the too diffuse thermocline, Zhang et al. (2022) using sensitivity experiments, showed that reducing the background diffusivity to better match with microstructure profiles, leads to significant improvements of the subsurface temperature in the equatorial Atlantic.

- In boreal summer and in the ATL3 region, the OMIP1 ensemble mean depicts a 51% greater interannual SST variability and a 33% larger temperature variability at the thermocline level compared to the OMIP2 ensemble mean (Figure 11a).

- In boreal summer, both OMIP ensembles exhibit a comparable magnitude of dT/dz in the ATL3 region (Figure 11b). This suggests that, relative to the OMIP2 ensemble, heightened interannual SST and temperature variability in the OMIP1 ensemble cannot be attributed to differences in the magnitude of dT/dz.

- Substituting the CORE-II winds for the JRA55-do winds in MOM5-LR, an OMIP2-like model, results in a 40% increase in boreal summer interannual SST variability and a 45% increase in temperature variability, as depicted in Figure 11a. This underscores the critical role of wind forcing in accurately simulating interannual SST variability in the EEA within ocean models. It's worth noting that, in comparison to MOM5-LR, the magnitude of dT/dz in MOM5-LR-winds remains unchanged (Figure 11b).

- In boreal summer, the equatorial Atlantic thermocline tilt within OMIP models varies between 24 m and 39 m, while it reaches 30 m in the case of ORA-S5 (Figure 11c). No correlation between the thermocline tilt and the ATL3 interannual SST variability is observed in OMIP models. It is worth noting that both MOM5-LR and MOM5-LR-winds exhibit a similar thermocline tilt, suggesting that the increased ATL3 interannual SST variability in MOM5-LR-winds is not attributable to a change in the thermocline tilt (Figure 11c).

- During AMJ, the zonal wind stress variability in the western equatorial Atlantic is slightly more pronounced in the OMIP1 ensemble mean compared to the OMIP2 ensemble mean. This difference may have played a role in the heightened interannual SST variability observed in ATL3 within the OMIP1 ensemble mean (as illustrated in Figure 11d). It is important to stress that the peak in ATL4 zonal wind variability occurs in April for the JRA55-do forcing and in May for the CORE-II forcing.





    – In boreal summer, the interannual SSH variability in the ATL3 region is 43% greater in the OMIP1 ensemble mean compared to the OMIP2 ensemble mean (Figure 11e). Sensitivity experiments reveal that this change in ATL3 interannual SSH variability from the OMIP1 ensemble to the OMIP2 ensemble is mainly attributed to the wind forcing. Furthermore, when comparing MOM5-LR to MOM5-LR-winds, the SSH variability in the ATL3 region during boreal summer is heightened by 13%, as depicted in Figures 10 and 11f.

In summary, this study has shown, by comparing the OMIP1 and OMIP2 ensembles and by using sensitivity experiments, that seemingly minor uncertainties in the atmospheric forcings can lead to notable discrepancies in the simulated interannual variability in the EEA region. For the equatorial Atlantic, we have shown that the interannual variability is particularly sensitive to the wind forcing in line with the results from Wen et al. (2017).

## 6.2 Discussion

It could be argued that changes in ocean model physics from OMIP1 to OMIP2 could also have lead to discrepancies in the simulation of the interannual variability in the equatorial Atlantic. However, models participating in both OMIPs have used the same ocean model physics. Hence, discrepancies in the interannual SST variability in the EEA should be rooted in the atmospheric forcing. The simulation of interannual variability in the EEA region by ocean models may be influenced by several factors other than the wind forcing. Beyond the zonal and meridional winds, the forcing from CORE-II and JRA55-do includes shortwave and longwave heat fluxes, precipitation, river runoff, air temperature at 2 m and evaporation.

The impact of shortwave and longwave heat flux forcing on interannual SST and temperature variability in the EEA has been examined using the MOM5-LR-heat sensitivity experiment. In MOM5-LR-heat, the JRA55-do heat fluxes were replaced by CORE-II heat fluxes during the last cycle. MOM5-LR-heat shows only slightly larger interannual SST and temperature variability in the ATL3 region compared to MOM5-LR (Figure S10). This suggests that shortwave and longwave heat flux forcing can only explain a small fraction of the increased interannual SST and temperature variability in the OMIP1 ensemble mean relative to the OMIP2 ensemble mean (Figure S10). However, the impact on the simulation of interannual temperature and SST variability in the EEA by precipitation, river runoff, 2 m air temperature and evaporation has not been investigated in this study.

The influence of the ocean horizontal resolution on the simulation of EEA interannual SST variability by ocean models under JRA55-do forcing has also been examined. Model pairs such as ACCESS-OM2 and ACCESS-OM2-025, MOM5-LR and MOM5-HR, as well as CMCC-CM2-HR4 and CMCC-CM2-SR5, were compared to each other. Each model pair has the same number of vertical levels but different horizontal resolution: 1° by 1° and 0.25° by 0.25°. This comparison, based only on three model pairs, suggests that increasing the ocean horizontal resolution does not lead to consistent changes in the equatorial Atlantic mean-state and interannual SST variability in boreal summer (Figure 11). One notable change is the increase of the vertical ocean temperature gradient in boreal summer when comparing MOM5-LR to MOM5-HR. However, this change is not observed in the other two model pairs.





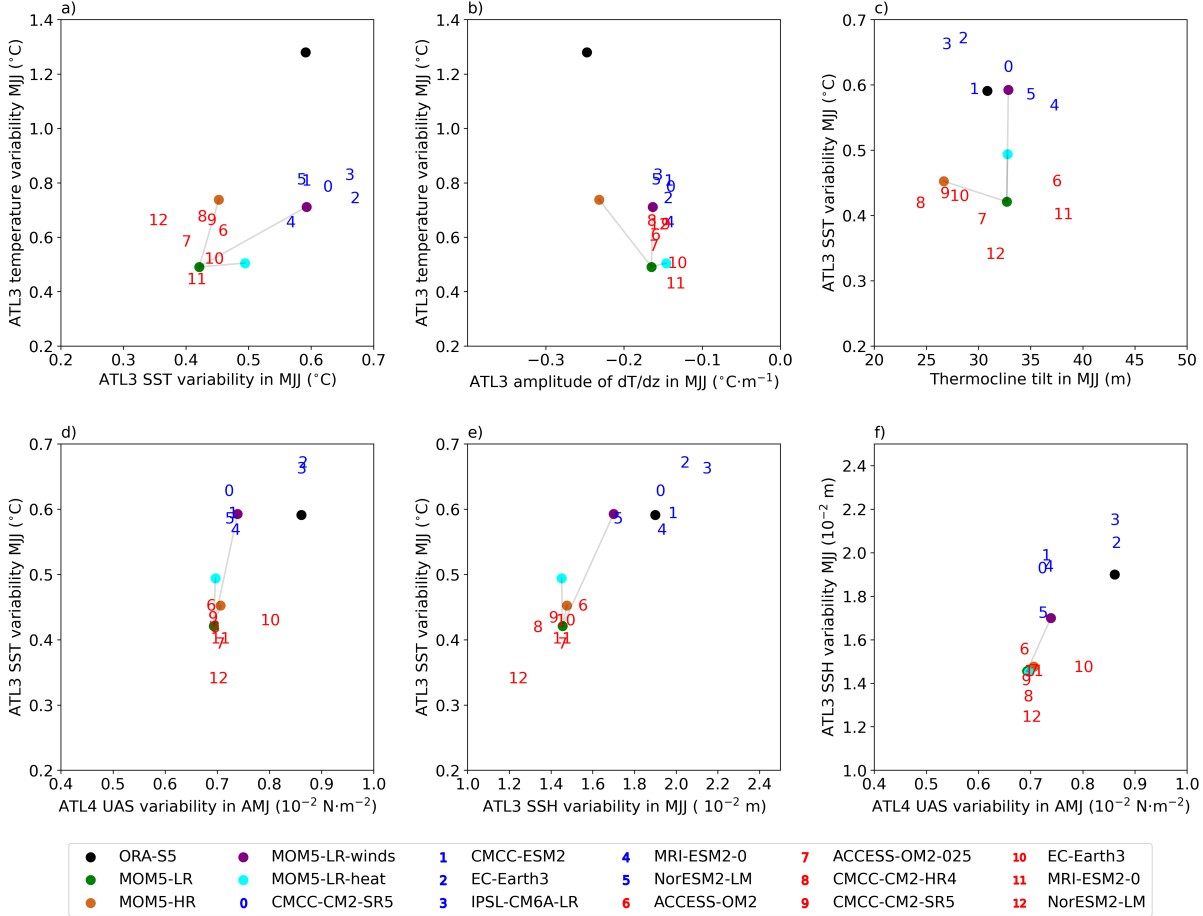

**Figure 11.** Scatter plots illustrating various equatorial Atlantic metrics assessed during the period January 1985 to December 2004. (a) Relationship between the standard deviation of ATL3-averaged SST anomalies in MJJ and the ATL3-averaged temperature anomalies in MJJ within ±10 m around the mean thermocline. (b) Relationship between ATL3-averaged dT/dz within ±10 m around the mean thermocline in MJJ and the ATL3-averaged temperature anomalies in MJJ within ±10 m around the mean thermocline. (c) Relationship between the equatorial Atlantic thermocline tilt in MJJ and the standard deviation of ATL3-averaged SST anomalies in MJJ. The equatorial thermocline tilt is defined as the difference between ATL4-averaged and ATL3-averaged depth of the maximum dT/dz. (d) Relationship between the standard deviation of ATL4-averaged UAS anomalies in MJJ and the standard deviation of ATL3-averaged SST anomalies in MJJ. (e) Relationship between the standard deviation of ATL3-averaged SSH anomalies in MJJ and the standard deviation of ATL3-averaged SST anomalies in MJJ. (f) Relationship between the standard deviation of ATL4-averaged UAS anomalies in MJJ and the standard deviation of ATL3-averaged SSH anomalies in MJJ. Dots are colour-coded: black, green, brown purple and cyan dots represent ORA-S5, MOM5-LR, MOM5-HR, MOM5-LR-winds and MOM5-LR-heat, respectively. Blue (red) numbers denote the OMIP1 (OMIP2) models.





Our findings underscore the importance of the wind forcing in modelling the interannual variability of the equatorial Atlantic.
Therefore, it is advisable to sustain and enhance wind observations in the tropical Atlantic in order to improve the quality of the reanalysis products. In addition, our results suggest that even though the seasonal cycle of the equatorial Atlantic winds is relatively well captured by reanalysis datasets, the interannual variability of the wind needs more validation in the tropical Atlantic. Taboada et al. (2019) conducted a comparative study of different reanalysis products and highlighted the lack of agreement among them in the tropics.

With respect to the CORE-II atmospheric state (Large and Yeager, 2009), the JRA55-do surface dataset (Tsujino et al., 2018) seem to improve the simulation of SST variability in extratropical areas and upwelling regions in OGCMs (Figure 1), likely due to its higher temporal and spatial resolution. However, the use of JRA55-do surface dataset results in weak interannual SST variability, not only in the equatorial Atlantic (Figure 6) but also in the equatorial Pacific (Figure 1). Further investigations are needed to analyse the changes in SST variability in the equatorial Pacific and their origins.

*Code availability.*

The MOM numerical ocean model version 5 is available from https://github.com/mom-ocean/MOM5. The scripts to reproduce the figures of the manuscript are available upon request to the corresponding author.

*Data availability.*

The ORA-S5 data is publicly available at the link: https://cds.climate.copernicus.eu/cdsapp#!/dataset/reanalysis-oras5?tab=
form. The OI-SST data is publicly available at the link: https://psl.noaa.gov/data/gridded/data.noaa.oisst.v2.html. The CCMP v2 data is publicly available at the following link: https://apdrc.soest.hawaii.edu/erddap/griddap/hawaii_soest_3387_f2e3_e359.html. The AVISO SLA was retrieved at the following link: https://cds.climate.copernicus.eu/cdsapp#!/dataset/10.24381/cds.4c328c78?tab=overview. The NCEP-R1 data is publicly available at the following link: https://psl.noaa.gov/data/gridded/data.ncep.reanalysis.html. The NCEP/DOE-R2 data is publicly available at the following link: https://psl.noaa.gov/data/gridded/
data.ncep.reanalysis2.html. The ERA5 data is available at the following link: https://cds.climate.copernicus.eu/cdsapp#!/dataset/reanalysis-era5-single-levels-monthly-means?tab=form.

The OMIP1 and OMIP2 model output data were downloaded at the following link: https://esgf-data.dkrz.de/projects/esgf-dkrz/. The CORE-II forcing is available at the following link: https://data1.gfdl.noaa.gov/nomads/forms/core/COREv2.html. The JRA55-do forcing is available at the following link: https://climate.mri-jma.go.jp/pub/ocean/JRA55-do/. The MOM5-LR,
MOM5-LR-winds, MOM5-LR-heat and MOM5-HR datasets used in this study can be retrieved from Farneti (2023).



*Author contributions.*

AP carried out the analyses and wrote the first draft of the manuscript. RF ran the sensitivity experiments and participated in the conceptualization, editing, and reviewing of the manuscript.

*Competing interests.*

The authors declare no competing interests.

*Acknowledgements.* We acknowledge the infrastructure and financial support of The Abdus Salam International Center for Theoretical Physics (ICTP). We also thank the climate modelling groups for producing and making available their model output, the Earth System Grid Federation (ESGF) for archiving the data and providing access.



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
