# Peer review of "An assessment of equatorial Atlantic interannual variability in OMIP simulations"

_EGUsphere, 2024_

## Author Comment (AC1)

**Response to reviewers' comments for "An assessment of equatorial Atlantic**

 **interannual variability in OMIP simulations".**

We thank the reviewer for their comments and suggestions that helped to improve the manuscript. Please find our detailed responses below. The reviewer comments are in black and our answers in blue. **When line numbers are given, they refer to the revised manuscript**

**with track changes accepted.**

This paper focuses on the evaluation of the realism of the seasonal and interannual variabilities in the Atlantic equatorial band (3°S-3°N) as simulated by some global ocean models in the context of the Ocean Model Intercomparison Project Phases 1 (OMIP1) and 2

(OMIP2). The two exercises differ in the surface forcing, i.e. CORE-II for OMPI1 and JRA55-do for OMIP2. Ensemble means are computed using 6 models for OMIP1 and 7 models for

OMIP2, and analyses are performed over a 20-year period (1985-2004). The authors report classical biases in the ocean mean state for OMIP1 and OMIP2 and highlight a drastically reduced interannual variability in OMIP2 (compared to OMIP1) in SSH, SST, and subsurface temperature. Using model experiments with the GFDL-MOM5 model, they attribute the differences between OMIP1 and OMIP2 interannual variability to surface wind forcing.

**General comments:**

This paper is useful for the modeling community and for the improvement of ocean models.

The figures are of good quality and the writing is good. However, the paper could be significantly improved. In particular:

- The introduction needs to be entirely revised. The actual introduction is based on the analysis of 4 figures (Figures 1 and 2, and Figures S1 and S2) that are already part of the paper's results. On the other hand, the forced and coupled dynamics of the equatorial Atlantic are hardly explained. One can also wonder why it is important to document the equatorial

Atlantic interannual variability. Specific questions seem to be thrown at the end of the introduction. 1) Why analyzing the seasonal cycle, knowing that the paper focuses on the interannual variability? 2) Analyzing the difference in interannual variability between OMIP1

and OMIP2: we already know that OMIP2 lacks variability, it has been diagnosed in Figures 2,

S1, and S2. 3) Does the interannual variability depend on the atmospheric forcing used?: This is a rhetorical question because OMIP1 and OMIP2 differ only in their atmospheric forcing (see your own comment at line 425).

We thank the reviewer for their suggestions to improve the introduction of our manuscript.

Following reviewer's comments, we have largely modified the introduction based on the following points:

● We have removed the paragraph referring to Figures 1, 2 and S1, S2. Figure 1 has been moved to the discussion section and the panels showing the OMIP1 and OMIP2

ensemble means of the standard deviation of the MJJ-averaged SSTA in Figure 2 have been removed. Figure S1 has also been removed.

● To motivate the study of the equatorial Atlantic variability, we have added to the introduction potential impacts of the equatorial Atlantic interannual variability on the onset of the West African Monsoon (L24-25), on El Niño/ Southern Oscillation, on the local chlorophyll-a concentration, on the Indian Monsoon and European climate. L42-

46.

● To motivate the analysis of the monthly climatology of zonal winds, SLAs and SSTs we have highlighted the strong link between the equatorial Atlantic monthly climatology and the equatorial Atlantic interannual variability as shown by Prodhomme et al.

(2019). L51-53

Prodhomme, C., Voldoire, A., Exarchou, E., Deppenmeier, A.-L., García-Serrano, J., and

Guemas, V.: How Does the Seasonal Cycle Control Equatorial Atlantic Interannual Variability?,

Geophysical         Research         Letters,         46,         916–922, https://doi.org/https://doi.org/10.1029/2018GL080837, 2019.

- It would be very nice if the authors could use the data from the PIRATA buoy network to assess the monthly climatological state of the ocean models. Depending on the availability of observations, the authors could also assess the realism of the interannual temperature variability in OMIP1 and OMIP2 using PIRATA data.

We agree with the reviewer that including the PIRATA data to the study would be very
informative. However, given our study period, 1985/01 to 2004/12, relatively little data is
available from the PIRATA. In numbers, the percentage of monthly mean zonal wind, dynamic
height and SST data available from the PIRATA moorings in the equatorial Atlantic over the
period 1985/01-2004/12 is provided in Table R1.

|        | Uwind at 4 m height | Dynamic height | SST |
|--------|---------------------|----------------|-----|
| 35˚W   | 83/240 ≈ 34.6%      | 83/240 ≈ 34.6% | 83/240 ≈ 34.6% |
| 23˚W   | 65/240 ≈ 27.1%      | 70/240 ≈ 29.2% | 70/240 ≈ 29.2% |
| 10˚W   | 88/240 ≈ 36.7%      | 88/240 ≈ 36.7% | 71/240 ≈ 29.6% |
| 0˚E    | 83/240 ≈ 34.6%      | 83/240 ≈ 34.6% | 83/240 ≈ 34.6% |

Table R1. Availability of zonal wind at 4 m height, dynamic height and SST at different mooring
sites over the period from January 1985 to December 2004.

The limited amount of available data over the period 1985/01-2004/12 is mainly due to the
fact that the PIRATA program started in the late-1990's. Yet, we have replicated Figure 2 from
the revised manuscript using the available data of zonal wind at 4 m height, dynamic height
and SST from the PIRATA buoy network, as depicted in Figure R1.

[Figure]

*Figure R1. Hovmöller diagrams of monthly climatologies for equatorial Atlantic U10, SLA, and*

*SST. (a) Monthly climatology of CCMP v2 U10, averaged between 1˚S and 1˚N, presented as a*

*function of longitude and calendar month for the period January 1987 to December 2004. (b,*

*c) Same as (a), but for CORE-II and JRA55-do U10 over the period January 1985 to December*

*2004. In (a, b, c) monthly climatologies derived using equatorial PIRATA mooring data at*

*35˚W, 23˚W, 10˚W, and 0˚E over the period from January 1985 to December 2004 are shown*

*by colored dots. (d) Monthly climatologies of the zonal wind at 35˚W, 0˚N and at 10m height*

*from CCMP v2 (orange), CORE-II (black), and JRA55-do (blue) and measured at 4 m height from*

*the 35˚W PIRATA mooring (purple). (e, f, g) Monthly climatologies of SLA in ORA-S5, OMIP1*
*ensemble mean, and OMIP2 ensemble mean, averaged between 1˚S and 1˚N, shown as a*
*function of the longitude and calendar month for the period from January 1985 to December*
*2004. In (e, f, g) monthly climatologies of dynamic height derived using equatorial PIRATA*
*mooring data at 35˚W, 23˚W, 10˚W, and 0˚E over the period from January 1985 to December*
*2004 are shown by colored dots. (h) Monthly climatologies of the SLA at 0˚E, 0˚N from ORA-*
*S5 (red), OMIP1 (black), OMIP2 (blue) and dynamic height from the 0˚E PIRATA mooring*
*(purple). (i, j, k) Same as (e, f, g) but for the SST. (l) Monthly climatologies of SST at 10˚W, 0˚N*
*from ORA-S5 (red), OMIP1 (black), OMIP2 (blue) and from the 10˚W PIRATA mooring of*
*(purple).*

Figures R1a-c show that the monthly climatology of zonal winds from CCMP-V2, CORE-II, and
JRA55-do in the equatorial Atlantic align closely with the PIRATA data in terms of phasing.
Figure R1d indicates that the zonal wind recorded at the 35˚W PIRATA mooring is generally
weaker compared to the reanalysis products throughout the year. This could be due to the
fact that PIRATA wind measurements are taken at 4 m height, while the reanalysis products
deliver data at 10 m height. Figures R1e-h depict that the OMIP1 and OMIP2 ensemble means
accurately capture both the phasing and amplitude of the monthly climatology of SLA in the
equatorial Atlantic. Similarly, Figures R1i-k illustrate that the phasing and amplitude of the
monthly climatology of SST in the equatorial Atlantic are well represented by ORA-S5, OMIP1,
and OMIP2 ensemble means. Finally, Figure R1l shows that the monthly climatology of SST
from OMIP1 and OMIP2 ensemble means at 10˚W, 0˚N closely resembles that from the
PIRATA mooring at 10˚W, however, with a warm bias.

We have included Figure R1, and its discussion, as Figure S10 in supplementary Text S2 of the
revised version.

- The model experiments carried out with the GFDL-MOM5 model (Section 5) are not very
informative, knowing that the seasonal and interannual variability in the equatorial Atlantic
is mostly linear. If the model uses classical bulk formulations (this information is not given in
the manuscript), then the prescribed surface winds control many aspects of the surface forcing (wind stress, latent, and sensible heat, evaporation). In particular, the model sensitivity experiment (MOM-LR-winds) designed to analyze the role of the surface winds on the interannual variability does not allow to disentangle the momentum forcing from the heat and freshwater forcing, which is a weakness for the interpretation of the results.

Furthermore, the use of bulk formulae to estimate the surface wind stress is accompanied by a drastic dependence of the wind stress amplitude on the climatological SST (see how the drag coefficient is estimated in the model), which again limits the interpretation of the difference between MOM-LR and MOM-LR-winds. An additional experiment could be run with the GFDL-MOM5 model to analyze the effect of changes in the mean state on the interannual variability. I suggest running MOM-LR forced by climatological winds / wind stress from CORE-II and the anomalies from JRA55-do. Or test the role of the forcing off the equatorial band, as compared to the local equatorial forcing.

The experiment MOM5-LR-winds was designed to test the sensitivity of the equatorial

Atlantic interannual variability to different wind forcing. Our aim, undoubtedly with a crude setting, was intentionally not to separate the effect of the prescribed wind on different surface forcing. As such, MOM5-LR-winds did provide a test for the difference in the equatorial Atlantic interannual variability between OMIP1 and OMIP2, which we concluded arises primarily from the wind forcing. However, following the reviewer's suggestion, we have replaced the MOM5-LR-winds experiment with a new experiment, MOM5-LR-anom, which is forced by climatological winds from JRA55-do and monthly anomalies from CORE-II.

Comparing MOM5-LR to MOM5-LR-anom in the revised manuscript has enabled us to observe more clearly the impact of the wind variability from the CORE-II forcing on the equatorial

Atlantic interannual variability.

- Note that the seasonal cycle is the seasonal deviation relative to the ocean mean state. For this study, the authors have to (estimate and) refer to the monthly climatology, which, in contrast, does contain the long-term mean.

We refer now to monthly climatology instead of seasonal cycle where applicable throughout the manuscript.

**Specific comments:**

1.  **Introduction:**

**Fig.1**: Caption mentions anomalies, are these interannual anomalies? If yes, improve the caption and clearly state that you are describing interannual variability in **Lines 17-20**. Note that in many studies such as in M. Martìn-Rey's work, they do not only remove the linear trend, but they remove the 7-yr low-frequency component (using fft).

In Figure 1 of the submitted manuscript the monthly mean SST anomalies without any filtering were considered. We have added "monthly mean" anomalies in the caption of Figure 10 of the revised manuscript. In Figure 10 we do not want to consider only the interannual SST variability as we also want to show the SST variability occurring at higher frequency like in eddy-rich regions like the Gulf Stream, Kuroshio, Malvinas and Agulhas currents as well as in eastern boundary upwelling systems.

**Fig.1**: The boxes can be removed. Also, NINO3.4 is not used in the article.

The boxes have been removed in Figure 10 of the revised manuscript (which was Figure 1 of the submitted manuscript).

**L27**: You could replace Dakar with Senegal to have two country names.

We have removed this sentence in the revised manuscript.

**L28**: "Discrepancies" should be replaced by differences.

This sentence has been removed in the revised manuscript.

**L45**: ENSO acronym has already been defined (and is used only twice in the paper).

We have removed the ENSO acronym as it was used only twice.

**L55**: There is an unnecessary closing bracket.

We thank the reviewer for spotting that. The extra closing bracket has been removed.

**L64**: "was comparable", do you mean that the magnitude was comparable?

Indeed, what is meant is that the magnitude of the ocean temperature variability was comparable. We have modified this sentence L60-62.

**L67**: OMIP acronym has already been defined.

**L70**: CMIP acronym has already been defined.

We thank the reviewer for spotting that. We made sure in the revised manuscript that the

OMIP and CMIP acronyms are defined only once.

1. **Data**

**Table 1**: The ocean resolution column is not a resolution but a number of points. What are the criteria that make you choose these specific models? Are these all available models with a resolution lower or equal to 1°x1°? Why did you choose an unequal number of models between OMIP1 and OMIP2? I notice that some of the models are identical between the two phases 0-9, 2-10, 4-11, and 5-12. Why can't you use the same model ensemble for both phases?

We have added in the caption that what is indicated in Table 1 of the revised manuscript is not the resolution but the number of grid points in the longitudinal, latitudinal and vertical dimensions.

As indicated in the submitted manuscript L122, only the models with a resolution finer that

1˚ by 1 ˚ are considered. However, we understand that the sentence is not precise enough, therefore we have rephrased it as follows: "All ocean models with a resolution finer than 1˚

by 1˚ and having all the variables needed for this study are listed in Table 1". L103-104 We realized that we were missing one model output that fits our criteria, MIROC6, which is now included in the OMIP1 and OMIP2 ensembles.

We could have used the same model ensemble for both phases, but we have decided to use the maximum number of models available. Considering only the model pairs would be interesting but it would be a limited subset of the total model data.

**L119**: The 55km zonal resolution is only the resolution at the equator.

We thank the reviewer for the precision, we have added: "at the equator" in the revised
manuscript L100.

L129: What is the criterium to choose 18 CMIP6 models?

The choice of these 18 CMIP6 models was not based on any particular criterium. In the revised
version we consider now all CMIP6 models available (55 models) on https://esgf-
data.dkrz.de/search/cmip6-dkrz/ having the variable TOS over the historical period from the
variant r1i1p1f1. Table S1 has been updated accordingly.

**L130**: What is rli1p1f1?

R1i1p1f1 is the variant reference. CMIP6 netCDF file metadata includes the variant-id global
attribute which has the format r1i1p1f1, where the numbers are indices for particular
configurations of:
- r: realisation (i.e. ensemble member)
- i: initialisation method
- p: physics
- f: forcing
More information can be found at: https://docs.google.com/document/d/1h0r8RZr_f3-
8egBMMh7aqLwy3snpD6_MrDz1q8n5XUk/edit?usp=sharing

**L133**: Add modeling to "We conducted several experiments".

"Modelling" has been added in L114.

**L134:** What does z* mean?

z* is the rescaled geopotential coordinate used by the model MOM for representing the free- surface (Adcroft and Campi, 2004; Griffies et al., 2016). For the large scale, z* surfaces differ slightly from constant geopotential surfaces z. We have modified the sentence L114-116.

Griffies, S. M., Danabasoglu, G., Durack, P. J., Adcroft, A. J., Balaji, V., Böning, C. W.,

Chassignet, E. P., Curchitser, E., Deshayes, J., Drange, H., Fox-Kemper, B., Gleckler, P. J.,

Gregory, J. M., Haak, H., Hallberg, R. W., Heimbach, P., Hewitt, H. T., Holland, D. M., Ilyina, T.,

Jungclaus, J. H., Komuro, Y., Krasting, J. P., Large, W. G., Marsland, S. J., Masina, S., McDougall,

T. J., Nurser, A. J. G., Orr, J. C., Pirani, A., Qiao, F., Stouffer, R. J., Taylor, K. E., Treguier, A. M.,

Tsujino, H., Uotila, P., Valdivieso, M., Wang, Q., Winton, M., and Yeager, S. G.: OMIP

contribution to CMIP6: experimental and diagnostic protocol for the physical component of the Ocean Model Intercomparison Project, Geoscientific Model Development, 9, 3231–3296, https://doi.org/10.5194/gmd-9-3231-2016, 2016.

Alistair Adcroft, Jean-Michel Campin: Rescaled height coordinates for accurate representation of free-surface flows in ocean circulation models, Ocean Modelling, Volume

7, Issues 3–4, 2004, Pages 269-284, ISSN 1463-5003, https://doi.org/10.1016/j.ocemod.2003.09.003.

**L136**: What does nominal mean?

We have removed "nominal" from the sentence.

**L140:** What is the bulk formula used for the estimation of momentum/heat/freshwater fluxes? What about the rivers, is there a relaxation to climatological SSS or runoffs?

Following the OMIP-CORE-II experimental protocol, our simulations make use of the Large and Yeager (2009) bulk formulae for computing turbulent fluxes. There is no restoring term applied to SST. A weak restoring to a monthly observational-based climatology is applied to sea surface salinity, as for all OMIP simulations, with a piston velocity of 50m/300d.

Large, W. G. and Yeager, S. G.: The global climatology of an interannually varying air–sea flux data set, Climate Dynamics, 33, 341–364, https://doi.org/10.1007/s00382-008-0441-3, 2009.

**L142:** Specify where the 10m-winds are used in the surface forcing estimation (wind stress, latent and sensible heat, evaporation).

In both OMIP1 and OMIP2 (see Large and Yeager, 2009; Griffies et al., 2009; Griffies et al.

2016), bulk formulae parameterize the turbulent fluxes of momentum, heat (sensible and latent), and moisture (evaporation) in terms of the near surface atmospheric state which includes the 10m winds. In the revised manuscript, when describing the new sensitivity experiment MOM5-LR-anom, we now clearly specify that the anomalous winds have an impact on all surface fluxes forcing the ocean. L126-130

Griffies, S. M., Biastoch, A., Böning, C., Bryan, F., Danabasoglu, G., Chassignet, E. P., England,

M. H., Gerdes, R., Haak, H., Hallberg, R. W., Hazeleger, W., Jungclaus, J., Large, W. G., Madec,

G., Pirani, A., Samuels, B. L., Scheinert, M., Gupta, A. S., Severijns, C. A., Simmons, H. L.,

Treguier, A. M., Winton, M., Yeager, S., and Yin, J.: Coordinated Ocean-ice Reference

Experiments        (COREs),        Ocean        Modelling,        26,        1–46, https://doi.org/https://doi.org/10.1016/j.ocemod.2008.08.007, 2009.

**L147:** Specify if the prescribed longwave is the longwave_in or the sum of longwave_in and longwave_out (that depends on SST**4).

Following both OMIP1 and OMIP2 protocols, the net surface longwave solar QL is computed from the downwelling longwave flux QA from the atmospheric state minus the blackbody radiation from the ocean back to the atmosphere, which depends on SST**4 (Large and

Yeager, 2008; Griffies et al., 2009). Given that we have now removed the discussion on the experiment MOM5-LR-heat and that the specifications of air-sea fluxes are part of the OMIP

protocol detailed in both Griffies et al. (2009) and Griffies et al. (2016), we have opted for not adding this information in the revised manuscript.

**L163:** Is this potential density?

Yes, it is potential density. We have added 'potential' to the sentence L149.

**L167**: What is the expected influence of a change in the thermocline tilt? Modal dispersion?

One could expect that with a greater thermocline tilt, the interannual SST variability in the
eastern equatorial Atlantic would be larger. Cai and Cowan (2013) showed for the Indian
Ocean dipole, using CMIP3 and CMIP5 models, that for a given wind anomaly, a greater
thermocline slope results in a stronger thermocline response, inducing a greater SST anomaly
in the eastern Indian ocean, than a weaker thermocline slope. They found that models with
greater climatological thermocline slope exhibit stronger thermocline feedback. However, as
discussed in this study, we find no relationship between the climatological thermocline tilt
and the interannual SST variability in the eastern equatorial Atlantic using the OMIP1 and
OMIP2 ensembles.

Cai, W., and T. Cowan (2013), Why is the amplitude of the Indian Ocean Dipole overly large in
CMIP3 and CMIP5 climate models? *Geophys. Res. Lett.*, 40, 1200–1205,
doi:10.1002/grl.50208.

L170: The feedbacks could be explained in the introduction, along with the impacts of changes
in certain components.

We have explained the different Bjerknes feedback components in the introduction L36-42,
but we kept the section 2.2.3 because we explain in that section that the different
components are obtained by linear regressions done in particular seasons and with particular
indexes.

1.  **Comparison of the monthly climatologies**

**L177**: See my general comment on the definition of seasonal cycle *vs.* monthly climatology.
Also introduce this section, because it is not obvious to all readers why it is important to
evaluate the realism of the ocean mean state and its seasonal variations and what are the
implications of biases on the interannual variability.

We have now indicated at the beginning of this section the following: "Accurately simulating the equatorial Atlantic wind, SLA and SST monthly climatologies in ocean models is crucial for the good representation of the EEA interannual SST variability (Prodhomme et al., 2019)." L163-164

**L185**: The seasonal cycle of SLA is driven by resonance modes (Brandt et al., 2016) associated with baroclinic modes 2 (at semiannual frequency) and 4 (at annual frequency).

Brandt, P., Claus, M., Greatbatch, R. J., Kopte, R., Toole, J. M., Johns, W. E., and Böning, C. W.: Annual and semiannual cycle of equatorial Atlantic circulation associated with basin-mode resonance, J. Phys. Oceanogr., 46, 3011–3029, https://doi.org/10.1175/Jpo-D-15-0248.1, 2016.

We thank the reviewer for the precision. We have now added this reference along with a sentence to the revised manuscript. L178-179

**L195**: Can you comment on the eastern part of the basin, which is more important for the Bjerknes feedbacks.

Following the reviewer's suggestion, we have now added the ATL3-averaged SLA in JJA for ORA-S5, OMIP1, and OMIP2 ensemble means in the paragraph and in Table 2 of the revised manuscript. L184-188

L214: What about the stratification (you could use the 24°C isotherm for the calculation).

We are not sure what the reviewer is suggesting with the 24°C isotherm. We agree with the reviewer that computing the stratification for each OMIP model could be interesting to compare, however, we believe that it would not provide more insight than the vertical temperature gradient. In addition, it would require to download the salinity field which represents a lot of data.

**L198-233**: Summarize all the estimated values in a table. The text is too technical to grasp the main message.

We have now included a table at the end of this section (Table 2 of the revised manuscript)

to summarize all values. We have also modified the text to read better.

**Figure 3**: On the right, you should add 3 curves for ATL4 or ATL3 averaged values. The y-axis labels should be centered between ticks positioned at the beginning and end of the month.

Currently, half a month is missing at the beginning of January and half a month is missing at the end of December. Furthermore, the figure caption can be reduced. Sentences are too repetitive.

As proposed by the reviewer, we have added on the right side of the revised Figures 2 and 4,

3 curves for ATL4 or ATL3 averaged values. We have also centered the y- axis ticks on the 15$^{th}$

of the month and the figure caption has been reduced.

**Figure 4**: For a better comparison, can you align subplot a) with subplots c) and g), and align subplot b), with subplots e) and i). Can you plot ORAS5 vertical velocity?

As suggested by the reviewer, all subfigures in the revised Figure 3 are aligned. Unfortunately,

ORA-S5 does not provide the vertical velocity, this is why it has not been plotted. In order to align the plots and because the vertical velocity from ORA-S5 is missing, we have decided to remove the subpanels (d, f, h, j) from Figure 3 of the revised manuscript.

1.  **Comparison of the interannual variability**

**L254**: Imbol Koungue et al (2017) is not an appropriate reference, this study is not about equatorial waves as it focuses on Benguela Niño/Niña events.

We have replaced this reference with Illig et al., (2004). L234-235

Illig, S., B. Dewitte, N. Ayoub, Y. du Penhoat, G. Reverdin, P. De Mey, F. Bonjean, and G. S. E.

Lagerloef (2004), Interannual long equatorial waves in the tropical Atlantic from a high- resolution ocean general circulation model experiment in 1981–2000, *J. Geophys. Res.*, 109,

C02022, doi:10.1029/2003JC001771.

**L259**: Why don't you compare OMIPs to ORAS5.

We have modified the sentence to: "The interannual SSH variability in the ATL3 region is too
strong (weak) in the OMIP1 (OMIP2) ensemble mean compared to ORA-S5 (Figure 4f, g, h). In
numbers, the OMIP1 (OMIP2) ensemble mean ATL3-averaged SSH variability in MJJ is 0.02 ±
0.002 m (0.015 ± 0.002 m), while it is 0.019 m in ORA-S5 (Figure 4h)." L240-242

**Figure 5**: On the right, you should add 3 curves for ATL4 or ATL3 averaged values. Caption:
What do the horizontal lines highlight? Specify that vertical lines denote the ATL4/ATL3
regions. The caption could be drastically reduced: "Same as Figure 3 but for the monthly
climatological standard deviation of interannual anomalies."

As proposed by the reviewer, we have added on the right side of the figure the 3 curves for
ATL4 or ATL3 averaged values. We have reduced the caption as suggested by the reviewer
and indicated what are the different vertical and horizontal lines.

**Figure 6**: Does BF1 have some meaning in the case of a forced simulation?

We thank for the reviewer for raising this point. Indeed, the western equatorial Atlantic zonal
wind response to an SST anomaly in the eastern equatorial Atlantic cannot be observed in a
forced ocean simulation. Therefore, we have removed the BF1 from Figure 5 of the revised
manuscript. We have added to the text the following: "The first component of the Bjerknes
feedback is not discussed as in a forced ocean model simulation there is no response of the
western equatorial Atlantic winds to an SST anomaly in the eastern equatorial Atlantic." L270-
272.

**L270**: Mention (here or in the introduction) that the peaks of variability correspond to the
classical Atlantic Niños/Niña events phase-locked to boreal spring/summer and the Atlantic
Niño II in November-December (Okumura and Xie, 2006).

Okumura, Y., and S. Xie, 2006: Some Overlooked Features of Tropical Atlantic Climate Leading
to    a    New    Niño-Like    Phenomenon.    J.    Climate,    19,    5859–5874,
https://doi.org/10.1175/JCLI3928.1.

We have added this reference along with a sentence to the introduction. L30-32

**L279**: Replace the word disparities with biases.

We have replaced the word "disparities" with "biases" as proposed by the reviewer. L261-
262

**L300**: I guess that the plus/minus 10 meters has been chosen quite arbitrarily?

Yes, the plus/minus 10 meters has been chosen arbitrarily as it encompasses the high
interannual temperature variability around the thermocline.

**L302**: How does the thermocline depth influence the MLD. The MLD is controlled by
momentum stress, isn't it?

We apologize for the confusion, the verb "influence" was badly chosen. What is meant is that
subsurface temperature anomalies at the thermocline level in the western equatorial Atlantic
are too deep to reach the MLD and hence they would not impact the temperature in the MLD.
L284-285

**L303**: The thermocline is not that close to the MLD, maybe the word "closer" is better here.

We agree with the reviewer and have replaced "close" by "closer". L286

**L307-L310**: Quantify by how much the subsurface temperature anomalies have been reduced
compared to ORAS5 (or from OMIP1 to OMIP2).

The subsurface temperature variability in MJJ in the ATL3 averaged between ±10 m around
the thermocline is of 1.28 ˚C for ORA-S5, 0.78 ± 0.06 ˚C for the OMIP1 ensemble mean and
0.58 ± 0.07 ˚C for the OMIP2 ensemble mean. Hence, relative to OMIP2, the equatorial
Atlantic Ocean interannual temperature variability in MJJ in the OMIP1 ensemble mean is
about 34% larger.  L290-292 and L299-302.

$(0.78 - 0.58)/0.58 \cong 0.345\%$.

**Figure 7**: On top of each panel, you could plot the interannual SSH variability (STD), which
should mirror the subsurface temperature variability.

As proposed by the reviewer, we have added the interannual SSH variability in MJJ on top of each equatorial Atlantic temperature variability section in MJJ of the revised Figures 6 and 7. As expected, the interannual SSH variability mirrors the subsurface temperature variability.

1. **Sensitivity tests on the wind forcing**

**L328-329**: In forced ocean models/simulations, the surface forcing controls the mean state, the seasonal cycle, and the variability. This statement is quite empty here (same as question 3 at the end of the introduction). What could be important to test is the effect of the forcing away from the Atlantic equatorial band as opposed to the local equatorial forcing.

We have removed this statement from the manuscript: "This underscores the sensitivity of the simulation of the tropical Atlantic interannual variability to surface forcings." We agree with the reviewer that testing the effect of the forcing away from the equatorial Atlantic band as opposed to the local equatorial forcing could be very interesting, however, we believe that it would fit better in a separate study.

**L334-351**: I do not see the purpose of comparing CORE-II and JRA55-do to other surface wind products. Can you please introduce this paragraph with your objectives?

The main objective of this paragraph is to show that quite some uncertainty exists among the atmospheric reanalysis products. We have removed this paragraph from the manuscript and included it into the supplementary material Text S3.

**L351**: Have these simulations/model configurations been validated?

We have added a validation of the mean-state, monthly climatology and interannual variability of the MOM5-LR and MOM5-HR simulations relative to ORA-S5 and we have compared MOM5-LR-anom to MOM5-LR. This can be found in the supplementary material Text S1.

**L352**: I do not get the implication of the "consequently".

We have removed "consequently" in the revised manuscript.

**L372**: "We have demonstrated" is a very strong statement. In the equatorial Atlantic, the
ocean dynamics is mostly linear (see work by P. Brandt, S. Illig, or others), so there is no
surprise here. That is why the shift of one month in the wind forcing causes the shift of one
month in SSH variability (Line 366-368).

We agree with the reviewer that "We have demonstrated" is a too strong statement. We have
replaced "demonstrated" by "shown" in the revised manuscript L338.

**Figure 9**: The subplots c) and d) are mistakenly referred to as a) and b).

We thank the reviewer for spotting this typo. It has been corrected in the revised manuscript.

**Figure 10**: On the right side of the plot, you should add ATL3 curves for both SSH (top panels)
and SST (bottom panels).

As proposed by the reviewer, we have added on the right side of the revised figure the ATL3-
averaged curves for both SSH and SST.

1. **Conclusion and Discussion**:

**L389-391**: this statement seems out of context.

This statement has been removed in the revised manuscript.

**L394-396:** This can be proven with model experiments (see my general comment)**.**

Please see our response above relative to the introduction of a new sensitivity experiment
MOM5-LR-anom.

**L407:** This can be associated with the estimation of the drag coefficient.

It is true that transfer coefficients for drag, sensible heat transfer and evaporation they all
have a dependency to the momentum flux. For this reason, the larger wind stress variability
in OMIP1 may play a dominant role in the strengthened SST variability in the ATL3 region.
Given that we have not investigated this aspect more in detail we prefer to leave this
statement as a suggestion.

**L424**: The fact that models in OMIP1 and OMIP2 use the same model physics should be said in section 2.1.2. This echoes with my previous question: why do you use different models for

OMIP1 and OMIP2 ensemble means?

We have added a sentence in section 2.1.2 indicating that models participating in both OMIPs use the same model physics. See also our response above regarding the choice of different

OMIP1 and OMIP2 models.

**L425:** see my general comment.

Please see our related responses above.

**Figure 11**: The point associated with MOM-LR-winds could be blue because it is closer to

OMIP1 protocol (shown with blue numbers).

We have changed the color for OMIP1 and OMIP2 models which are now in black and blue, respectively. Because MOM5-LR is a OMIP2-like simulation and MOM5-LR-anom is a OMIP1- like simulation we put them in blue and black in Figure 9 of the revised manuscript.

---

## Author Comment (AC2)

**Response to reviewers' comments for "An assessment of equatorial Atlantic interannual variability in OMIP simulations".**

We thank the reviewer for their comments and suggestions that helped to improve the manuscript. Please find our detailed responses below. The reviewer comments are in black and our answers in blue. **When line numbers are given, they refer to the revised manuscript with track changes accepted.**

This study compares tropical Atlantic variability among forced ocean simulations (CORE-I and CORE-II) and a subset of CMIP6 models and identifies a diffusive thermocline bias among models.

My primary concern with this study is that the model representation is biased towards Eulerian vertical coordinate models such as MOM5 . NorESM is the only isopycnal coordinate configuration , however it is using a high background vertical diffusivity ( nominally 1-1.5e-5 m2 s-1). Near-equatorial background levels are reduced in several CMIP configurations, notably NOAA/GFDL-CM2G (https://doi.org/10.1175/2008JPO3708.1) ,which is a quasi-Isopycnal coordinate model, similar to NorESM.

We agree with the reviewer that the study is biased towards Eulerian vertical coordinate models. However, we have used all available models participating to OMIP phases 1 and 2 with a resolution higher than 1° by 1° and presenting all the variables needed for our analysis (L102-103). The NOAA/GFDL-CM2G is a coupled model and therefore is not participating to the Ocean Model intercomparison Project.

Echoing the reviewer's concern, we believe that diversity among ocean models should be encouraged, whereas we observe instead a global convergence towards a handful of global ocean models, often using similar numerical approaches and parameterizations. Hence, more isopycnal coordinate models, or models using generalized vertical coordinates and the vertical Lagrangian-remap method (Griffies et al., 2020), contributing to OMIPs and CMIPs would be beneficial for both model development and assessment.

We have added a note on this topic in the Discussion section (L400-405).

Griffies, S. M., Adcroft, A., & Hallberg, R. W. (2020). A primer on the vertical Lagrangian-
remap method in ocean models based on finite volume generalized vertical
coordinates. *Journal of Advances in Modeling Earth Systems*, 12,
e2019MS001954. https://doi.org/10.1029/2019MS001954

Model sensitivity results suggest that increasing model resolution slightly reduces the diffuse
thermocline bias (MOM5-HR).  This is not discussed further and deserves further attention.
Would an implication be that additional high resolution studies are needed to assess to what
degree stratification bias can be reduced by increasing horizontal resolution?  To what extent
could improved representation result from numerics (e.g. Lagrangian coordinate
models)?  Including an isopycnal with low equatorial diffusivities (CM2G) would help to
address this question.

We agree with the reviewer that this topic deserves more attention. As mentioned in the
manuscript, we have 3 model pairs ACCESS-OM2 and ACCESS-OM2-025, MOM5-LR and
MOM5-HR, as well as CMCC-CM2-HR4 and CMCC-CM2-SR5 which have the same number of
vertical levels but they differ in their horizontal resolution, going from coarse ($1°×1°$) to
refined ($0.25°×0.25°$). This comparison, based only on three model pairs, suggests that
increasing the ocean horizontal resolution does not lead to consistent changes in the
equatorial Atlantic mean-state and interannual SST variability in boreal summer (Figure 9 of
the revised manuscript). One notable change is the increase of the vertical ocean temperature
gradient and subsurface temperature variability in boreal summer when comparing MOM5-
LR to MOM5-HR. However, this change is not observed in the other two model pairs. A larger
number of model pairs would be required to properly assess the impact of resolution. (L393-
400)

Furthermore, Zhang et al., (2022) investigated the impact of the wind forcing and ocean
vertical mixing parametrization on the tropical Atlantic subsurface ocean temperature bias in
the tropical Atlantic using sensitivity experiments made with the POP2 model. They found
that the wind forcing has only a marginal effect on the subsurface temperature bias in the tropical Atlantic. However, they showed that the overestimated vertical mixing in OGCMs play a major role in the formation of subsurface warm biases in the tropical Atlantic.

As mentioned above, comparing Eulerian versus Lagrangian coordinate models would help to shed light on this aspect, but it is not presently feasible with the available OMIP simulations.

Zhang, Q., Y. Zhu, and R. Zhang, 2022: Subsurface Warm Biases in the Tropical Atlantic and

Their Attributions to the Role of Wind Forcing and Ocean Vertical Mixing. *J. Climate*, **35**, 2291–

2303, https://doi.org/10.1175/JCLI-D-21-0779.1.

Figure quality is good. In Figures 3 and 4 (and perhaps 5), it would be helpful to show anomalies for all fields, with respect to ORA-S5.

We thank the reviewer for the appreciation of our figures. In the revised manuscript, we do not show the anomalies for all fields with respect to ORA-S5, as we think it is important for readers to properly see the phasing of each variable. Nonetheless, we have added, as suggested by reviewer 1, the ATL3 or ATL4 indexes for each variable in Figures 2, 4, and 8 of the revised manuscript, allowing for direct comparison. In addition, supplementary Text S1 is devoted to the comparison of the MOM5 model runs, MOM5-LR and MOM5-HR, to ORA-S5.

Did the authors consider analyzing mean and time-varying contributions to the upwelling heat budget, i.e. how much of the variability is related to changes in the background stratification/upwelling versus eddy contributions? This could be helpful for the disussion, however, the existing figures reasonably convey the point of the dominance of vertical processes in this region.

We thank the reviewer for the suggestion. A comprehensive heat budget analysis will be performed in a future study using only one model at varying resolution, and performing multiple sensitivity runs to investigate the role of the background stratification on the variability of the equatorial Atlantic.

---

## Author Response (AR2)

**Response to reviewers' comments for "An assessment of equatorial Atlantic interannual variability in OMIP simulations".**

We thank the Reviewer for their positive appreciation of our manuscript.

A few typos were corrected in the manuscript:

- In the table 3 of the revised manuscript, the OMIP2 ensemble equatorial thermocline tilt in MAM should be 35.44 ± 5.61 m and not 35.44 ± 3.52, this has been corrected.
- In the caption of Figure 2, "January 1987" has been corrected to "January 1988".
- The reference to Figure S3e in L248 was incorrect and it has been replaced with Figure S2e.
- "As previously discussed, we also find increased interannual SST variability in MOM5-LR-anom (Figure 8d) relative to MOM5-LR (Figure 8e)" L335-336 has been corrected to "As previously discussed, we also find increased interannual SST variability in MOM5-LR-anom (Figure 8e) relative to MOM5-LR (Figure 8d)" in the revised manuscript.
- In the caption of Figure 9d, "MJJ" was corrected to "AMJ" as indicated by the Figure 9d label.

Please find our detailed responses below. The Reviewer comments are in black and our answers in blue. **When line numbers are given, they refer to the revised manuscript with track changes accepted.**

Following my suggestions and comments, the authors have significantly improved the manuscript. I think the manuscript could be published as is, but I have some recommendations that I would like to suggest to the authors:

1) I appreciate the effort the authors put into the development of the new sensitivity experiment (MOM5-LR-anom). Nevertheless, I think that this new experiment should be analyzed in comparison to the previous sensitivity experiment the authors conducted, i.e., MOM5-LR-wind. In my opinion, the revised manuscript fails to draw stronger conclusions on the origin of the difference in interannual variability between OMIP1 and OMIP2. The weak difference in interannual variability between MOM5-LR-wind and MOM5-LR-anom suggests that it is controlled by the interannual anomalies in the wind, rather than the total wind, depreciating the role of the climatological forcing.

We appreciate the Reviewer's suggestion. As indicated by the Reviewer in the previous round of review, the MOM5-LR-wind sensitivity experiment was not ideal because of its crude setup.

In the revised manuscript, we have replaced it with MOM5-LR-anom, following the reviewer's recommendation. Comparing MOM5-LR and MOM5-LR-anom clearly reveals that the greater interannual variability in SST and SSH in MOM5-LR-anom is due to the interannual variability in the CORE-II wind forcing. A comparison between MOM5-LR-anom and MOM5-LR-wind would illustrate the impact of CORE-II interannual anomalies versus the total CORE-II winds on interannual SSH and SST variability in the equatorial Atlantic. As the Reviewer noted, this impact is minor and would necessitate additional figures and analysis. Therefore, we have decided not to reintroduce the MOM5-LR-wind experiment into the revised manuscript.

2) I would appreciate if the authors could provide a small table summarizing the experiments:

name, associated wind forcing, and heat/water/river forcing. Something likes:

OMIP1 – COAREII - COAREII

OMP2 – JRA-55 -JRA55

MOM5-LR – JRA55-JRA55

MOM5-LR-wind – COAREII – JRA55

MOM5-LR-anom – COAREIIclim+ JRA55anom – JRA55

We thank the Reviewer for the suggestion. We have included Table 2 in the revised manuscript, which summarizes the various GFDL-MOM5 simulations used in this study.

Specific comments:

L95: Add a reference for the AVISO SSH product

We were unable to find a reference paper corresponding to the vDT2021 SLA product. The only citation we found and used is: Copernicus Climate Change Service, Climate Data Store, (2018): Sea level gridded data from satellite observations for the global ocean from 1993 to present. Copernicus Climate Change Service (C3S) Climate Data Store (CDS). DOI:
10.24381/cds.4c328c78.

L111-112: The 1°x1° interpolation is already mentioned

Yes, this has already been mentioned for the OMIP models. However, we also indicate here
that the CMIP6 models were also interpolated on a 1˚ by 1 ˚ regular grid. We have added
"CMIP6" to the sentence. L113

Section 2.1.4: add the ref the Large and Yeager in this paragraph.

We have already cited this study in the previous section where we describe the OMIP models
and the CORE-II forcing.

L175: is it a seasonal cycle rather than monthly averages here?

That is correct, we have replaced "monthly climatology" by "seasonal cycle". L176 and 178

L244: the figures show high correlation, instead of exhibit

Corrected as suggested by the Reviewer. L245

L365: Your results underscore the role of wind interannual anomalies, rather than total wind
(see my first general comment).

We agree with the Reviewer and have revised the sentence. It reads now: "This underscores
the critical role of interannual anomalies in the wind forcing in accurately simulating the
equatorial Atlantic interannual variability within ocean models." L376-377

Section 6.1. I think that you could consider reordering the point stressed in the conclusions.
Wind could be discussed before SST and SSH?

We have reordered some of the key points in section 6.1. We now discuss the results of the
sensitivity experiments in the last two points.